# Global transcriptional analysis identifies a novel role for SOX4 in tumor-induced angiogenesis

Stephin J Vervoort[1†], Olivier G de Jong[2†], M Guy Roukens[1†], Cynthia L Frederiks[1†], Jeroen F Vermeulen[3], Ana Rita Lourenço[1], Laura Bella[4], Ana Tufegdzic Vidakovic[5], José L Sandoval[5], Cathy Moelans[3], Miranda van Amersfoort[3], Margaret J Dallman[6], Alejandra Bruna[5], Carlos Caldas[5], Edward Nieuwenhuis[7], Elsken van der Wall[8], Patrick Derksen[3], Paul van Diest[3], Marianne C Verhaar[2], Eric W-F Lam[4], Michal Mokry[7], Paul J Coffer[1,7]*

[1]Department of Cell Biology, Center for Molecular Medicine, University Medical Center Utrecht, Utrecht, The Netherlands; [2]Department of Nephrology and Hypertension, University Medical Center Utrecht, Utrecht, The Netherlands; [3]Department of Pathology, University Medical Center Utrecht, Utrecht, The Netherlands; [4]Department of Surgery and Cancer, Imperial Centre for Translational and Experimental Medicine, Imperial College London, Hammersmith Hospital Campus, London, United Kingdom; [5]Cancer Research UK Cambridge Institute, Li Ka Shing Centre, Cambridge, United Kingdom; [6]Department of Life Sciences, Division of Cell and Molecular Biology, Imperial College London, London, United Kingdom; [7]Division of Pediatrics, Wilhelmina Children's Hospital, University Medical Center Utrecht, Utrecht, The Netherlands; [8]Cancer Center, University Medical Center Utrecht, Utrecht, The Netherlands

*For correspondence:
p.j.coffer@umcutrecht.nl

[†]These authors contributed equally to this work

Competing interests: The authors declare that no competing interests exist.

**Abstract** The expression of the transcription factor *SOX4* is increased in many human cancers, however, the pro-oncogenic capacity of SOX4 can vary greatly depending on the type of tumor. Both the contextual nature and the mechanisms underlying the pro-oncogenic SOX4 response remain unexplored. Here, we demonstrate that in mammary tumorigenesis, the SOX4 transcriptional network is dictated by the epigenome and is enriched for pro-angiogenic processes. We show that SOX4 directly regulates endothelin-1 (ET-1) expression and can thereby promote tumor-induced angiogenesis both in vitro and in vivo. Furthermore, in breast tumors, SOX4 expression correlates with blood vessel density and size, and predicts poor-prognosis in patients with breast cancer. Our data provide novel mechanistic insights into context-dependent SOX4 target gene selection, and uncover a novel pro-oncogenic role for this transcription factor in promoting tumor-induced angiogenesis. These findings establish a key role for SOX4 in promoting metastasis through exploiting diverse pro-tumorigenic pathways.
DOI: https://doi.org/10.7554/eLife.27706.001

## Introduction

Transcription factor-mediated control of gene expression networks is crucial during embryonic development and thereafter for tissue homeostasis. Consequently, dysregulation of transcription factor function has been observed to result in a wide-variety of developmental defects and is often responsible for the pathogenesis of disease. A restricted subset of transcription factors is often

expressed in human cancers, aberrantly re-activating developmental pathways which then contribute to tumor-progression and metastasis (*Darnell, 2002*).

SOX4 is a prominent cancer-associated transcription factor and its mRNA expression is elevated in a large number of human cancers, being part of a general human cancer-associated gene expression signature (*Rhodes et al., 2004*). It is a member of the SRY-related HMG-box (SOX) family of transcription factors that have important roles in embryonic development and tissue homeostasis, for example by controlling differentiation and maintaining adult tissue stem cell populations (*Lourenço and Coffer, 2017*; *Vervoort et al., 2013a*; *Sarkar and Hochedlinger, 2013*). Sox4-deficient mice die during embryogenesis due to defective cardio-vasculature development (*Schilham et al., 1996*). In addition, SOX4 controls the development of specific tissues, including lymphoid, pancreatic, brain and bone (*Vervoort et al., 2013a*). In contrast to its wide-spread expression during embryogenesis, the expression of SOX4 in adult tissues is mostly restricted to adult stem- and progenitor cell populations including intestinal and hematopoietic stem cells (*Sarkar and Hochedlinger, 2013*; *Vervoort et al., 2013a*).

*SOX4* expression in human cancers has been positively correlated with tumor-progression in a wide-variety of solid and hematopoietic tumors (*Lourenço and Coffer, 2017*; *Vervoort et al., 2013a*). Accordingly, SOX4 hypomorphic mice have decreased cancer-incidence and a resistance to carcinogen-induced skin cancer (*Foronda et al., 2014*).

The pro-oncogenic function of SOX4 has been attributed to a number of key cell-intrinsic processes including cell proliferation, cell-cycle regulation and tumor stemness (*Vervoort et al., 2013a*). A recurring theme is that SOX4 endows tumor cells with a more migratory and invasive phenotype. This has been shown using in vitro assays employing a large variety of different tumor types, such as breast cancer (*Tavazoie et al., 2008*; *Zhang et al., 2012*), hepatocellular carcinoma (*Liao et al., 2008*), ovarian cancer (*Yeh et al., 2013*), prostate cancer (*Wang et al., 2013*) and lung cancer (*Zhou et al., 2015*). Moreover, SOX4 expression correlates with increased depth of invasion in clinical specimens (*Fang et al., 2012*; *Lin et al., 2013*). For a limited number of tumor types, downstream targets of SOX4 have been identified that were important for invasion such as NRP1 and SEMA3C (hepatocellular carcinoma; *Liao et al., 2008*), TEAD2 and RBP1 (lung cancer; *Castillo et al., 2012*) and EGFR, Tenascin C (prostate cancer; *Scharer et al., 2009*). However, despite the similarity in phenotype that SOX4 confers in the various cell types, the overlap of transcriptional targets in the different studies has proven to be very limited (*Vervoort et al., 2013a*) suggesting that SOX4 has context-dependent effects on tumor development.

A number of studies have indicated a role for SOX4 in mammary tumor progression. In breast cancer, *SOX4* is directly controlled by miRNA-335, the loss of which is associated with disease progression and poor metastasis-free survival (*Tavazoie et al., 2008*). *SOX4* has also been demonstrated to be a part of gene signatures associated with metastasis of breast tumors to the brain and lungs (*Minn et al., 2005*; *Bos et al., 2009*). Moreover, SOX4 has been shown to control the TGF-β-induced epithelial-to-mesenchymal transition (EMT), a process associated with increases in tumor-initiating cells, in invasive and migratory capacity, in metastasis and in drug resistance (*Zhang et al., 2012*; *Tiwari et al., 2013*; *Kalluri, 2009*; *Vervoort et al., 2013b*; *Lourenço and Coffer, 2017*). In mice, SOX4-mediated induction of EMT has been proposed to be mediated by direct regulation of the histone-lysine N-methyltransferase *EZH2*, resulting in alterations in local histone three lysine (K) 27 trimethyl (H3K27me3) occupancy followed by changes in expression of EMT-associated genes (*Tiwari et al., 2013*). In line with its increased metastasis-associated expression and role in EMT, depletion of *SOX4* has also been shown to reduce metastasis formation in mouse models of breast cancer (*Tiwari et al., 2013*; *Tavazoie et al., 2008*). The context-dependent nature of the SOX4 transcriptional response can in part be explained by cell-type-specific expression of cooperating transcription factors (*Wilson and Koopman, 2002*). Recently, it has been demonstrated in murine pancreatic ductal adenocarcinoma (PDA) that, the cell-type-specific expression of KLF5 can redirect the SOX4-mediated TGF-β transcriptional network from a pro-apoptotic to a pro-tumorigenic response, thus preventing lethal EMT (*David et al., 2016*). However, the wider relevance of this process in tumor progression remains to be elucidated.

Despite these observations, the transcriptional network and molecular mechanisms through which SOX4 contributes to the pathogenesis of breast cancer remain largely unknown. Moreover, SOX4 protein expression in both primary and metastatic tumor-tissue of patients with breast cancer has not been systematically studied. Here, by using an integrative genome-wide analysis approach, we

provide novel insight into the mechanisms determining SOX4 target gene selection. We define a 'core' SOX4-transcriptional network in mammary epithelial cells that reveals an unexpected role of SOX4 in tumor-induced angiogenesis through direct regulation of Endothelin-1 (ET-1). Our in vitro and in vivo experiments demonstrate a novel pro-oncogenic function of SOX4 in mammary tumor development as a promoter of tumor-induced angiogenesis, and furthermore we show that SOX4-high breast tumors are highly vascularized, aggressive and therapy-resistant. These findings provide both a resource for understanding SOX4 function as well as defining a novel role for this transcription factor in regulating tumor-induced angiogenesis.

## Results

### SOX4 is an epigenome-guided transcriptional activator

To better understand the functional role of SOX4 in tumorigenesis, it is imperative to understand how target gene specificity and activation are achieved. We aimed to characterize SOX4 chromatin-binding on a genome-wide level by using chromatin-immunoprecipitation and sequencing (ChIP-seq). To identify an antibody suitable for ChIP-seq, we first analyzed the ability of a number of SOX4 antibodies to immunoprecipitate (IP) SOX4 in combination with Rapid Immunoprecipitation Mass spectrometry of Endogenous proteins (RIME) (*Figure 1—figure supplement 1A*; *Figure 1—source data 1*) (*Mohammed et al., 2013*). This analysis yielded a single suitable antibody (Diagenode, CS-129–100), which was subsequently used for ChIP-seq analyses.

To globally evaluate SOX4 transcriptional function, the well-characterized human mammary epithelial cell line (HMLE) was utilized as a model system (*Taube et al., 2010*). We have previously demonstrated that HMLE cells express low but detectable levels of SOX4 and rapidly gain metastatic traits associated with EMT upon conditional activation of SOX4 (*Vervoort et al., 2013b*). In order to investigate the transcriptional output of SOX4, a doxycycline (DOX)-inducible SOX4 HMLE cell line (HMLE-S4) was generated, allowing rapid and conditional expression of SOX4 (*Figure 1—figure supplement 1B*). SOX4 ChIP-seq was subsequently performed in wild-type HMLE cells and in doxycycline-treated or -untreated HMLE-S4 cells. We simultaneously generated genome-wide binding profiles for RNA-polymerase II (POL2), the transcription start site (TSS)-associated histone mark histone H3 lysine (K) four trimethylation (H3K4me3), the repressive mark H3K27me3 and the active regulatory mark H3K27 acetyl (H3K27ac), in the latter (*Figure 1—source data 2*).

SOX4-bound loci were successfully identified in both HMLE wild-type (WT), untreated and DOX-treated HMLE-S4 cells and, as expected, the largest number of peaks was uniquely bound in DOX-treated HMLE-S4 cells (*Figure 1A*), as reflected by an increased SOX4 occupancy on the HDAC2 promoter (*Figure 1—figure supplement 1C*). To analyze the genomic distribution of SOX4 binding, we determined the distance of SOX4 peaks to the closest TSS. In HMLE WT cells, untreated and DOX-treated HMLE-S4 cells, SOX4 binding was enriched at TSS-proximal sites, compared to random genomic regions (*Figure 1B* and *Figure 1—figure supplement 1D–E*), indicating that SOX4 binding is enriched at promoter regions.

To both validate and gain further insight into the sequence requirements for SOX4 binding, de novo motif analysis was performed. As expected, the SOX4-consensus DNA-binding motif was the most highly enriched motif in DOX-treated HMLE-S4 cells (*Figure 1C*) and was present in over 67% of all peaks. The SOX4 consensus motif was also centrally enriched in the identified peaks, further confirming sequence-dependent binding (*Figure 1D*). In addition to the SOX4 motif, a number of co-occurring motifs were identified corresponding to AP-1, CTCF, ETS1 and SMAD3 (*Figure 1C*).

To further define the features that underlie binding of SOX4 to chromatin, the epigenetic profile of SOX4-bound sites in HMLE-S4 cells was analyzed. SOX4-binding sites were centrally enriched in histone marks associated with active and open chromatin (H3K4me3 and H3K27ac), as well as in high levels of POL2, whereas the repressive mark H3K27me3 was absent from SOX4-bound sites (*Figure 1E*). To quantify these effects, the discriminating features between SOX4-bound and -unbound sites within regions of open chromatin, as defined by H3K27ac signal, were assessed. Quantitative analysis of SOX4-bound versus -unbound H3K27ac sites revealed that SOX4-bound sites have significantly higher occupancy for H3K27ac, H3K4me3 and POL2 compared to unbound sites, and are devoid of H3K27me3 occupancy (*Figure 1F*). These findings indicate that SOX4 preferentially binds to active/open chromatin.

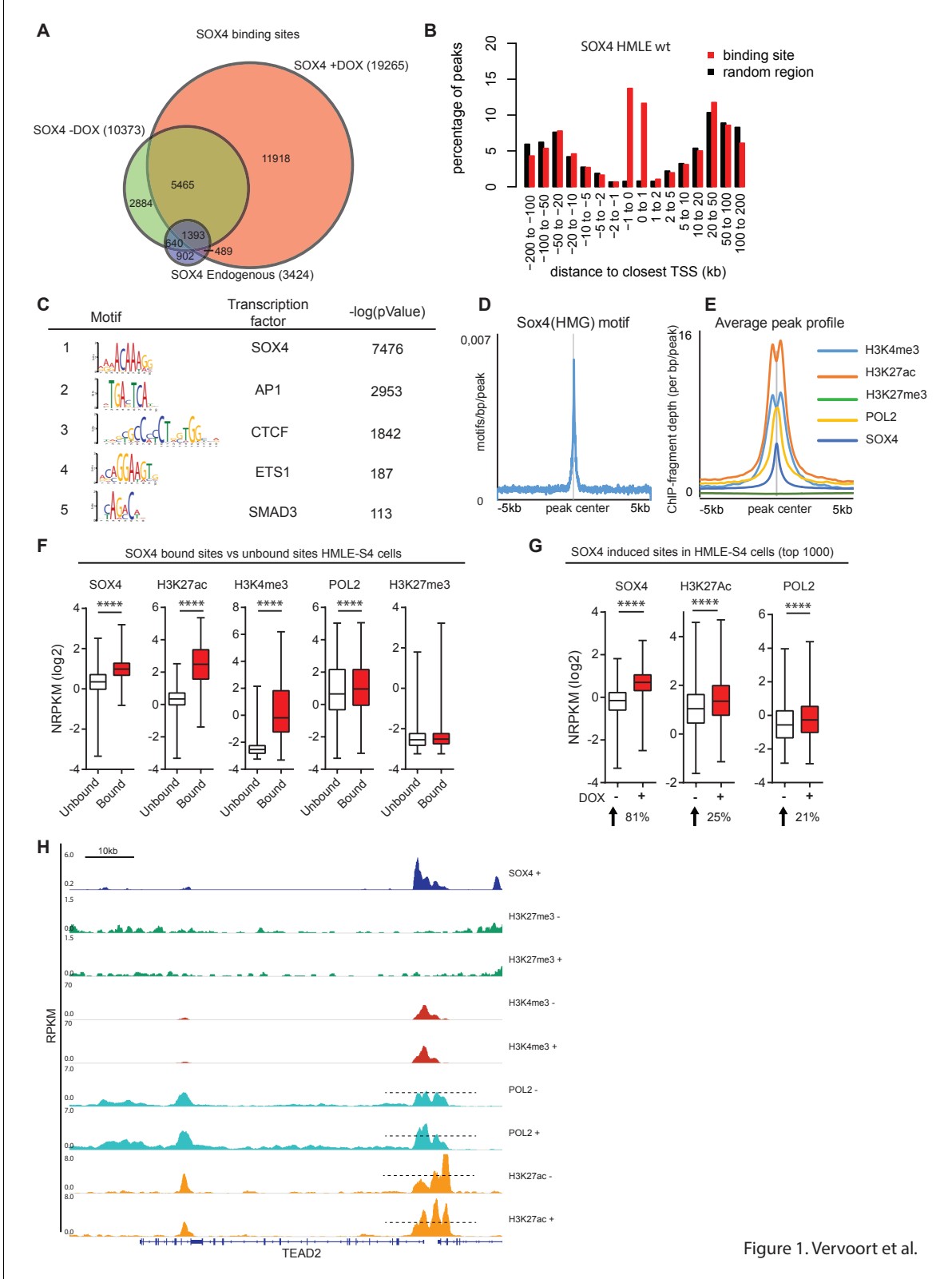

**Figure 1.** SOX4 is a transcriptional activator, preferentially binding to open/active chromatin. (**A**) Venn-diagram showing the overlap of SOX4 binding sites identified by ChIP-seq in WT-HMLE and DOX-induced HMLE-S4 cells. (**B**). Genomic distribution of SOX4 binding sites in WT-HMLE cells with respect to annotated genes, represented as distance from the nearest TSS. Random genomic regions were used as background. (**C**). De novo motif analysis of SOX4-bound sites in DOX-treated HMLE-S4 cells. Selected significantly enriched motifs are represented. SOX4 was identified as the most

*Figure 1 continued on next page*

*Figure 1 continued*

highly enriched motif. (**D**) Density of the consensus SOX4 motif in SOX4 bound sites in HMLE-S4 cells 500 bp up- and downstream of the peak center. (**E**) Occupancy plots of SOX4, H3K27ac, H3K27me3, POL2 and H3K4me3 in the 10kb-genomic region surrounding the SOX4 peak center. (**F**) Changes in SOX4, H3K27ac, H3K4me3, POL2 and H3K27me3 are shown for SOX4-bound and SOX4-unbound sites (5%–95% whiskers, two-tailed Mann-Whitney U test, **** indicate p<0.0001). (**G**) Changes in SOX4, H3K27ac, and POL2 in HMLE-S4 for top 1000 DOX-induced sites ranked by SOX4-signal (****p<0.0001, Wilcoxon-signed-rank test) (**H**) Genomic tracks representing the occupancy of SOX4, H3K27me3, H3K4me3, H3K27ac and POL2 in untreated and DOX-treated HMLE-S4 cells surrounding the genomic locus of the canonical SOX4 target gene *TEAD2*.

DOI: https://doi.org/10.7554/eLife.27706.002

The following source data and figure supplement are available for figure 1:

**Source data 1.** Analyses of Rapid Immunoprecipitation for Mass spectrometry on Endogenous proteins (RIME) to identify an optimaal anti-SOX4 antibody for ChIP described in *Figure 1A*.

DOI: https://doi.org/10.7554/eLife.27706.004

**Source data 2.** Overview of ChIPseq and RNAseq experiments described in *Figures 1* and *2*.

DOI: https://doi.org/10.7554/eLife.27706.005

**Figure supplement 1.** Antibody validation and ChIP-seq analysis.

DOI: https://doi.org/10.7554/eLife.27706.003

To assess whether SOX4-induction in HMLE-S4 cells can itself result in alterations in active histone marks and POL2, all H3K27ac loci were ranked according to their relative change in SOX4-signal, or H3K27ac-signal in DOX-treated versus -untreated HMLE-S4 cells. This revealed that induction of SOX4 results in a concomitant increase in POL2 and H3K27ac occupancy (*Figure 1G*), which is also apparent at the genomic locus surrounding the validated SOX4 target gene TEAD2 (*Figure 1H*) (*Bhattaram et al., 2010*). No change was observed for the subset of sites where SOX4 binding remains unaltered (*Figure 1—figure supplement 1F*). In agreement, the most highly induced H3K27ac sites showed increased SOX4 and POL2 occupancy (*Figure 1—figure supplement 1G*). Taken together, these findings show that increased binding of SOX4 to regions of open chromatin promotes an active chromatin-state.

The reliance on a pre-existing chromatin-state for SOX4 binding implies that SOX4 transcriptional effects are likely to be highly context dependent. Indeed, analysis of overlap between SOX4-bound sites in HMLE cells and sites identified in two distinct breast cancer cell lines, the HER2-amplified HCC1954 cells and highly metastatic triple-negative MDA-MB-231 cells, revealed a minor degree of overlap in SOX4-bound loci, although the overlap on the associated genes was higher (*Figure 1—figure supplement 1H–I*). SOX4 thus utilizes a pre-existing chromatin-landscape by binding to open/active chromatin, and binding to these sites subsequently increases the occupancy of active histone-marks and POL2.

## Characterization of the SOX4 transcriptional network

To characterize the SOX4 transcriptional network, RNA-sequencing (RNA-seq) was subsequently performed. To this end, HMLE cells stably expressing SOX4 fused to the estrogen-receptor hormone-binding domain (ERSOX4 HMLE) were utilized, allowing for extremely rapid activation of SOX4 by 4-hydroxytamoxifen (4-OHT) (*Vervoort et al., 2013b*). ER-control HMLE and ERSOX4 HMLE cells were stimulated with 4-OHT (8 hr) after which total RNA was isolated and subjected to RNA-seq to identify alterations in gene expression. Data analysis revealed that 1378 genes were significantly regulated (q-value < 0.05) and differentially expressed by at least two-fold upon conditional SOX4 activation (*Figure 2A*), whereas no significantly altered genes were identified in ER HMLE control cells.

To identify SOX4 direct transcriptional targets, the RNA-seq dataset was overlapped with genes containing a SOX4 -binding site within 20 kb of their TSS (*Figure 2B*). This revealed a core set of 650 genes, which were shown to be bound and transcriptionally regulated by SOX4 (*Figure 2B–C*). Notably, the largest fraction of genes showed increased expression upon SOX4 activation (63%), while a considerably smaller percentage were found to be repressed targets (37%).

In order to independently validate the targets identified in our RNA-seq dataset, quantitative real-time PCR (qRT-PCR) was performed on ER-control and ERSOX4 HMLE cells stimulated with 4-OHT (8 hr and 24 hr). For all selected target genes, SOX4 conditional activation resulted in a time-dependent increase in expression, with no significant change in the expression in ER-control HMLE

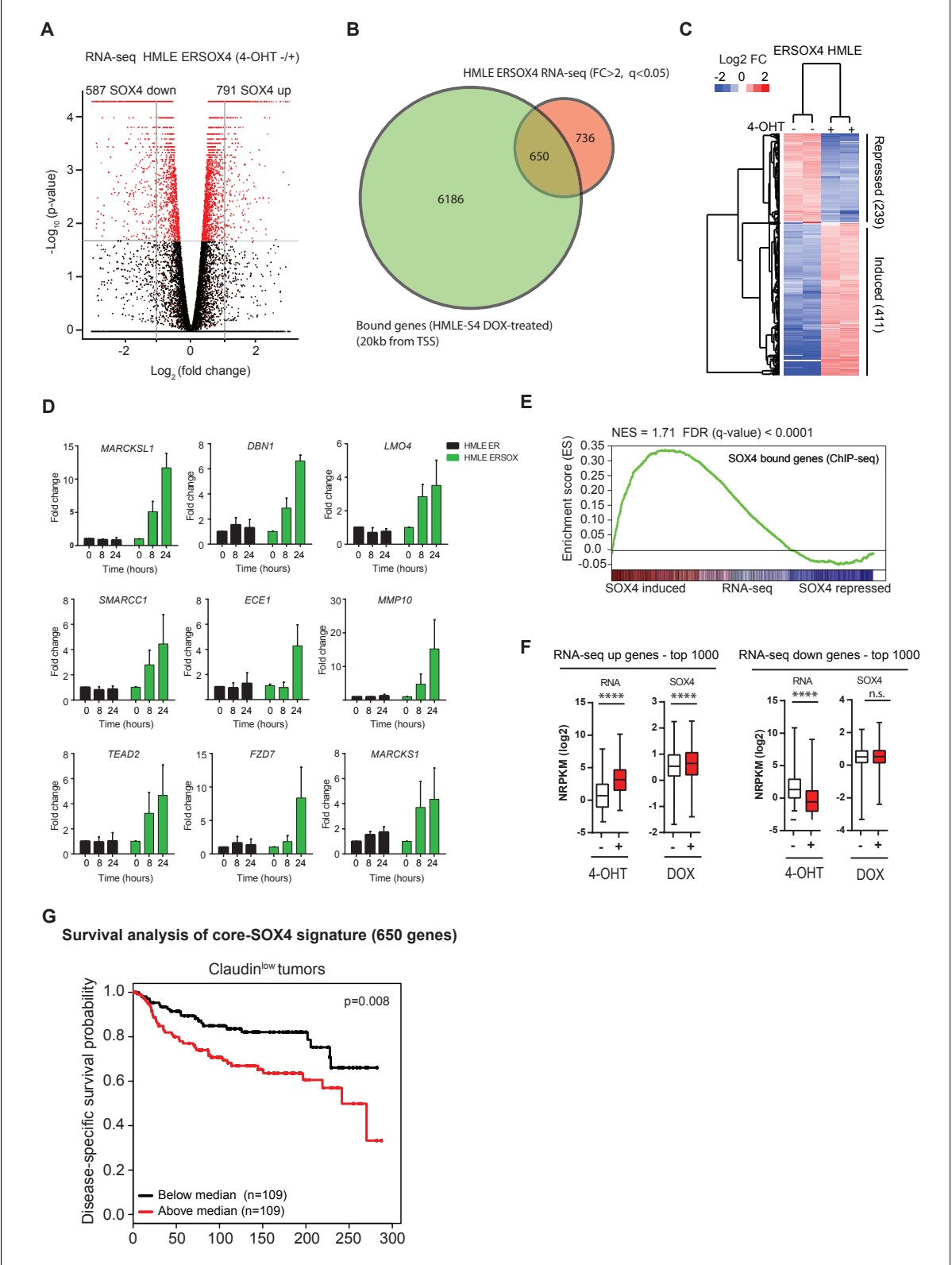

**Figure 2.** Identification of a novel core-SOX4 target gene signature with prognostic value and pro-angiogenic networks. (**A**) Gene expression analysis of SOX4 induced changes in HMLE ERSOX4 cells conditionally activated by 4-OHT for 8 hr and analyzed by RNA-seq. No significantly changed genes were observed in ER-control HMLE cells (data not shown). Significantly changed genes (q-value < 0.05) are indicated in red. (**B**) Venn diagram showing the overlap of genes regulated by SOX4 on the RNA-level (FC > 2, q-value < 0.05) with genes possessing a SOX4 binding site within 20 kb from an

*Figure 2 continued*

annotated TSS in DOX-treated HMLE-S4 cells (650 genes). (C). Heatmap visualizing gene expression changes for core-SOX4 target genes. (D) Quantitative real-time PCR analysis of gene expression changes in 4-OHT treated ER-control and ERSOX4 HMLE cells for selected SOX4 target genes identified by RNA-seq. Data represented as mean ± SD, normalized for *β2M* of three independent biological replicates. (E) Gene-set enrichment analysis (GSEA) representing the enrichment of SOX4 bound sites (2 kb from TSS) in the RNA-seq expression dataset ranked by log2 fold change upon conditional activation of SOX4. (F) Quantitative analysis of changes in SOX4 occupancy in peaks associated with most highly induced or repressed genes derived from the RNA-seq dataset generated in ERSOX4 HMLE cells (****p<0.0001, Wilcoxon-signed-rank test). (G) Analysis of the prognostic value of the core-SOX4 gene expression signature (650 genes) in claudin[low] breast cancer patients (METABRIC).

DOI: https://doi.org/10.7554/eLife.27706.006

The following figure supplement is available for figure 2:

**Figure supplement 1.** Overlap of ChIP-seq and RNA-seq datasets.

DOI: https://doi.org/10.7554/eLife.27706.007

cells (*Figure 2D*). These observations validate the RNA-seq and ChIP-seq results as shown for the MARCKSL1 locus (*Figure 2—figure supplement 1A*).

In support of the notion that SOX4 predominantly acts as a transcriptional activator, gene set enrichment analysis (GSEA), probing for enrichment of SOX4-bound genes (ChIP-seq, 2 kb from TSS) in the RNA-seq dataset demonstrated a significant and positive enrichment of SOX4-bound genes (*Figure 2E*). Furthermore, only positively regulated SOX4 target genes were significantly enriched for SOX4 binding as determined by hypergeometric testing (*Figure 2—figure supplement 1B*). Finally, a significant increase in SOX4 occupancy upon DOX-treatment of HMLE-S4 cells was observed with peaks associated with the top 1000 induced genes (*Figure 2F*), whereas no significant change was observed for peaks associated with repressed genes (*Figure 2—figure supplement 1C*). Taken together, these findings strongly indicate that SOX4 is a transcriptional activator and suggest that inhibitory effects on gene expression are the result of indirect secondary mechanisms.

In agreement with the observation that SOX4 transcriptional networks are context-dependent and directed in part by the epigenome, we observed that SOX4-target genes strongly correlated with decreased survival only in Claudin[low] breast cancers (*Figure 2G*). In contrast, only a moderate correlation with survival was observed in Luminal and normal-like breast cancers and no significant correlation was observed in Basal-like breast cancers (data not shown). These findings suggest that SOX4 transcriptional networks may be breast cancer subtype specific.

## SOX4 positively regulates *EDN1* expression and promotes ET-1 secretion

To identify enriched cellular processes in the core-SOX4 gene set (650 genes), gene-ontology (GO) analysis was performed. GO-term analysis of SOX4 core genes showed a significant association with blood vessel morphogenesis, in addition to cell cycle- and cell migration-associated processes (*Figure 3A*). SOX4 target genes were also enriched for genes encoding secreted proteins that are involved in cell migration, extra-cellular composition, angiogenesis and inflammation (*Figure 3—figure supplement 1A–B*). The association with vascular development and cell migration was restricted to the positively regulated gene-set, whereas terms related to cell cycle and apoptosis were enriched in repressed genes (*Figure 3—figure supplement 1C–D*).

A role for SOX4 in tumor-induced angiogenesis remains unexplored and could be an essential component of its pro-tumorigenic effects. To this end, we examined the genes associated with angiogenesis more closely, and 13 genes were found to be induced upon conditional SOX4 activation (*Figure 3B*). We chose to further evaluate the role of Endothelin-1 (*EDN1*) since its expression has been demonstrated to positively correlate with breast cancer tumor-grade and distant metastasis, and has been suggested to promote tumor-induced angiogenesis, migration and invasion (*Salani et al., 2000*; *Wülfing et al., 2004*). *EDN1* encodes preproendothelin, which through intracellular cleavage by convertases and Endothelin Converting Enzyme-1 (ECE-1) is converted into a 21-amino acid mature form (*Rosanò et al., 2013*). Endothelin-1 (ET-1) directly induces angiogenesis by binding to its cognate receptors (ETAR and ETBR) expressed on endothelial cells (*Rosanò et al., 2013*). Interestingly, both *EDN1* and its converting enzyme *ECE-1* were observed to be transcriptionally induced, indicating that SOX4 targets dual aspects of this pro-angiogenic pathway.

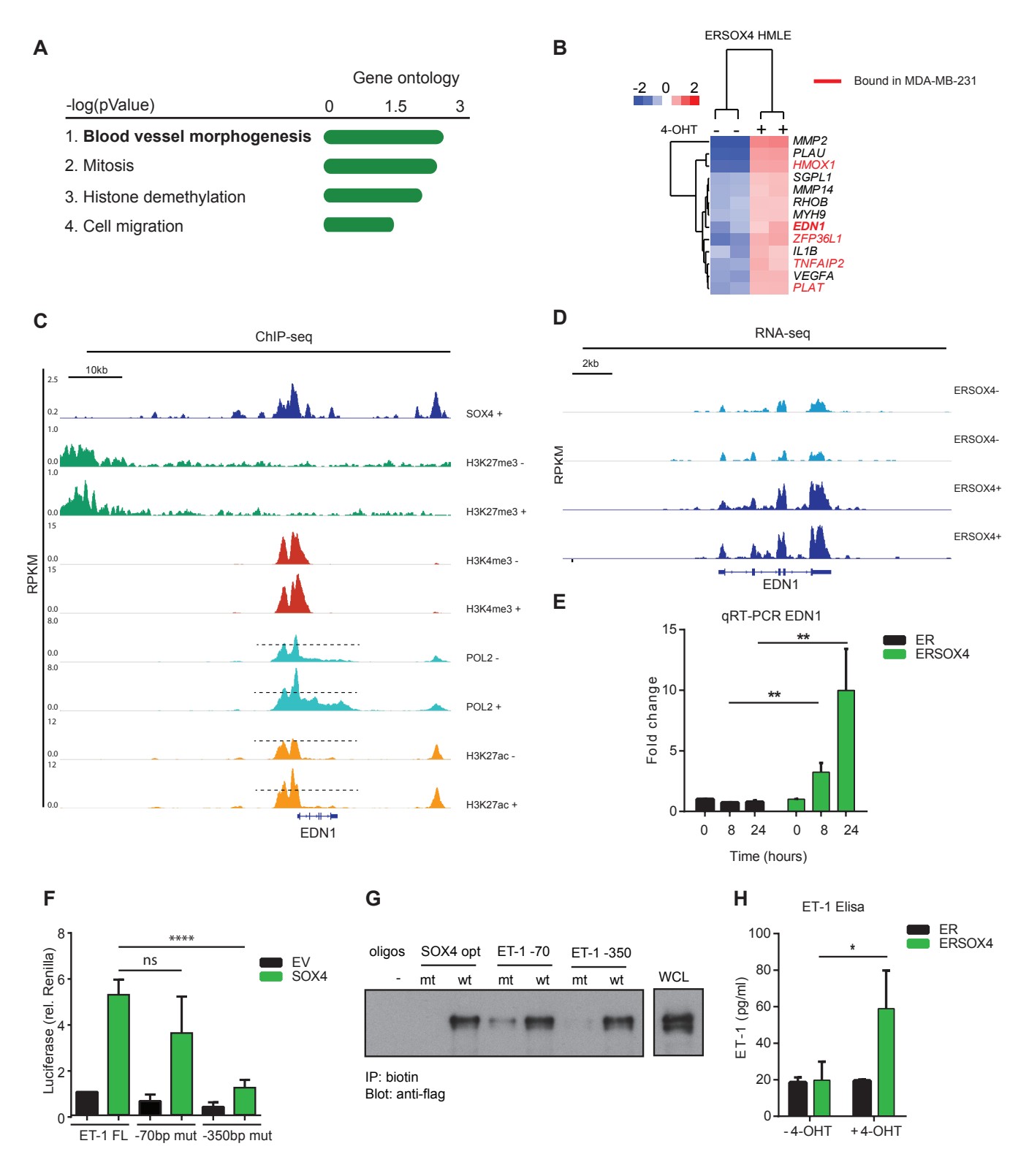

**Figure 3.** SOX4 directly regulates *EDN1* mRNA expression and promotes production of the mature ET-1 peptide. (**A**) Gene Ontology (GO) analysis on core-SOX4 target genes. Top terms are shown. (**B**) Heatmap of gene expression changes upon conditional SOX4 activation in HMLE cells for genes associated with the GO-term Blood vessel morphogenesis (HMLE regulated and bound). Genes with a SOX4-binding site within 20 kb of a TSS in MDA-MB-231 cells are annotated in red (**C**) Normalized ChIP-seq tracks for SOX4, H3K27me3, H3K4me3, POL2, and H3K27ac surrounding the *EDN1* locus in

*Figure 3 continued on next page*

*Figure 3 continued*

HMLE-S4 cells. For histone marks and POL2 both DOX-treated and -untreated profiles are represented (-/+). (D) Normalized RNA-seq profile for the *EDN1* gene in untreated and 4-OHT-treated ERSOX4 HMLE cells (-/+). (E) qRT-PCR analysis of *EDN1* expression in 4-OHT-treated ER and ERSOX4 HMLE cells. Data represented as mean ± SD, normalized to *β2M* of three independent biological replicates (**p-value<0.01, Student's t-test). (F) Luciferase assay in HEK293T cells transiently transfected with the full-length EDN-1 promoter (FL) as well as luciferase constructs in which the conserved sites have been mutated (−70 and −350). Results obtained from three independent biological replicates wherein three independent technical replicates were used per condition (ns: non-significant; ****p-value<0.001, Student's t-test). (G) Pulldown assay in Flag-Sox4 transfected HEK293T cells using biotinylated DNA-probes. Probes matching an optimal SOX4-binding sequence, and the −70 bp and −350 bp of the *EDN1* promotor sequence are indicated and mutated versions thereof (- indicates empty beads). Representative data obtained from three independent biological replicates. (H) Enzyme-linked immunosorbent assay (ELISA) analysis of ET-1 expression in the overnight conditioned media of 4-OHT treated and untreated HMLE-S4 cells. Results obtained from three independent biological replicates (*p-value<0.05, Student's t-test).
DOI: https://doi.org/10.7554/eLife.27706.008

The following figure supplements are available for figure 3:

**Figure supplement 1.** Analysis of patient survival and cellular processes associated with the core-SOX4 target gene signature.
DOI: https://doi.org/10.7554/eLife.27706.009

**Figure supplement 2.** SOX4 activates the *EDN1*-luciferase promoter and *SOX4* expression correlates with *EDN1* in breast cancer patients.
DOI: https://doi.org/10.7554/eLife.27706.010

In line with our previous observations, analysis of ChIP-seq and RNA-seq profiles of *EDN1* locus indicated a strong increase in POL2 and H3K27ac upon DOX-mediated activation of SOX4 in HMLE-S4 cells (*Figure 3C*) and an increase in mRNA expression of *EDN1* upon conditional activation of SOX4 in ERSOX4 HMLE cells (*Figure 3D*). This induction of *EDN1* expression was also observed by qRT-PCR in ERSOX4 HMLE cells, and was absent in control ER HMLE cells (*Figure 3E*). *EDN1* was also identified as a direct SOX4 target by ChIP in both MDA-MB-231 and HCC1954 cell lines (*Figure 1—figure supplement 1H* and data not shown). Indeed upon shRNA-mediated knockdown of SOX4 in these two cell lines we also observed that *EDN1* expression was decreased (Figure 6, Figure 6—figure supplement 1A–B for MDA-MB-231 and *Figure 3—figure supplement 2C* for HCC1954).

Closer examination of the *EDN1* promoter overlapping SOX4 and H3K27ac profiles identified two conserved SOX4 binding sites at 70 bp and 350 bp from the TSS (*Figure 3—figure supplement 2A*). To assess whether the SOX4 binding region (−961 bp - + 3 bp) upstream of the TSS is sufficient for transcriptional activation of EDN1 by this factor, luciferase assays were performed using a reporter construct containing the *EDN1* promoter region. Ectopic expression of SOX4 strongly induced luciferase expression as compared to the empty vector (EV) control, which could be inhibited by the co-transfection of a dominant negative truncated SOX4 lacking the transactivation domain (*Figure 3—figure supplement 2B*). Activation of the luciferase reporter was significantly inhibited when the SOX4 binding site at −350 bp was mutated, whereas no significant reduction was observed upon mutation of the −70 bp site, indicating that transcriptional activation of *EDN1* by SOX4 under these conditions is mediated by the site at −350 bp from the TSS (*Figure 3F*). To determine whether SOX4 can indeed directly bind these sites, pull-down assays were performed with biotinylated DNA-probes containing the identified *EDN1* promoter binding sites or a consensus SOX4 binding site (positive control) and mutated versions thereof. SOX4 was found to specifically bind to the conserved motifs at −350 bp from the *EDN1* TSS, which could be inhibited by mutation of this site (*Figure 3G*).

To assess whether transcriptional induction of *EDN1* by SOX4 also results in increased production of the mature ET-1 peptide, enzyme-linked immunosorbent assays (ELISA) were performed. Conditional activation of SOX4 in HMLE resulted in a significant increase in ET-1 secretion by these cells (*Figure 3H*). Taken together these observations demonstrate that SOX4 directly induces *EDN1* expression on the mRNA level resulting in increased expression of the secreted ET-1 peptide.

## SOX4 induced ET-1 expression promotes tumor-induced angiogenesis in vitro

Since activation of SOX4 resulted in increased expression of ET-1, we next investigated whether this was able to induce pro-angiogenic effects on endothelial cells. To this end, SOX4 in ERSOX4 HMLE cells was activated for a period of eight hours, and subsequently the media was refreshed for overnight conditioning. The SOX4-activated conditioned media (SCM+) was subsequently tested for its

ability to affect pro-angiogenic processes in human microvascular endothelial cells (HMEC-1) in three distinct in vitro assays assessing cell migration, network formation, and sprouting. In parallel, we investigated the pro-angiogenic function of synthetic ET-1 in these systems to ascertain whether this could phenocopy any effects observed in SCM+.

The functional effects of SCM+ on HMEC-1 cells were first investigated in a wound-healing assay (*van Balkom et al., 2013*). Relative to basal media alone, exposure of HMEC-1 cells to SCM+ was capable of increasing migration of HMEC-1 cells, whereas no significant increase was observed with SCM- or ER-control HMLE conditioned media (ERCM+) (*Figure 4*). Importantly, SCM+-induced migration could be inhibited by the dual endothelin receptor antagonist Bosentan (*Clozel et al., 1994*), demonstrating that the pro-migratory effect indeed required endothelin receptor activation (*Figure 4B*). SCM+ did not affect HMEC-1 cell proliferation, indicating that wound-closure is exclusively dependent on cell migration (*Figure 4—figure supplement 1A*). Similar to SCM+, treatment of HMEC-1 cells with a synthetic ET-1 peptide resulted in a significant increase in wound closure, although not to the same extent as complete media which contains additional pro-migratory and pro-angiogenic factors (*Figure 4C*). In order to assess whether the SCM+-mediated effects required transcriptional induction of *EDN1* by SOX4, siRNA-mediated knockdown of *EDN1* was performed in HMLE cells 24 hr prior to conditional SOX4 activation. *EDN1* knockdown reduced the SCM+-induced migration in HMEC-1 cells (*Figure 4D*, *Figure 4—figure supplement 1B*). Similar to conditional activation in HMLE cells, stable expression of SOX4 in MDA-MB-231 cells also resulted in increased *EDN1* expression (*Figure 4—figure supplement 1C*) and conditioned media with the capacity to increase HMEC-1 migration (*Figure 4E*).

To further evaluate the pro-angiogenic effects of ET-1 and SCM+, a matrigel angiogenesis assay was utilized. When plated in low-density on a matrigel surface, HMEC-1 cells rapidly form a meshed network, the length of which is an in vitro measure for their angiogenic potential. Relative to basal media alone, SCM+ treatment of HMEC-1 cells resulted in an increase in the relative network length, which could be inhibited by Bosentan, whereas no increase in network formation was observed in control conditions (*Figure 4—figure supplement 1D–E*). Similarly, incubation with the ET-1 peptide significantly increased the network length, an effect that could also be inhibited by treatment with Bosentan (*Figure 4—figure supplement 1F*).

In an analogous manner, the capacity of SCM+ and ET-1 to induce *in vitro* angiogenic sprouting was also investigated. Collagen-coated sephadex beads were incubated with HMEC-1 cells, allowing cell adhesion to the beads, after which cell-bound beads were embedded in matrigel. In the presence of pro-angiogenic stimuli, HMEC-1 cells form angiogenic sprouts that penetrate the matrigel layer. In this assay both ET-1 and SCM+ treatment of HMEC-1 cells resulted in a marked increase in sprouting, although less pronounced than full medium (*Figure 4F–J*). Relative to basal media, SCM + potently induced sprouting in this assay by approximately 6-fold which could be inhibited by Bosentan (*Figure 4F–G*). Correspondingly, ET-1 induced sprouting by approximately 5-fold, which could also be completely inhibited by Bosentan (*Figure 4H*). Similar to the results obtained in the wound-closure assays, siRNA mediated depletion of *EDN1* prior to conditional SOX4 activation resulted in an almost complete inhibition of SCM+-induced sprouting (*Figure 4I*). Additionally, conditioned media from SOX4 expressing MDA-MB-231 cells also resulted in increased HMEC-1 sprouting (*Figure 4J*). In accordance, shRNA-mediated knockdown of *SOX4* in MDA-MB-231 cells resulted in a decrease of *EDN1* expression (Figure 6A–B, Figure 6—figure supplement 1A–B) and a concomitant decrease in the ability to induce migration and sprouting of endothelial cells (*Figure 4—figure supplement 1G–H*).

Taken together, these findings show that SOX4 activation in HMLE and in MDA-MB-231 cells can bestow pro-angiogenic effects that are dependent on ET-1.

## SOX4 expression in breast epithelial cells induces angiogenesis in vivo in a zebrafish tumor-xenograft model

To determine whether SOX4 may regulate angiogenesis in vivo, we initially made use of a zebrafish tumor-xenograft model (*Figure 5A*). In this model, fluorescently labeled tumor cells (red) are injected into the yolk sac of 1 day post-fertilization (dpf) zebrafish embryos in proximity to the subintestinal vein (SIV) structure. In the presence of tumors, alterations in the normal development of the SIV, such as protrusions extending from the structure, are considered to be representative of tumor-induced angiogenesis. Ectopic neovascularization originating from the subintestinal vessels (SIV) can

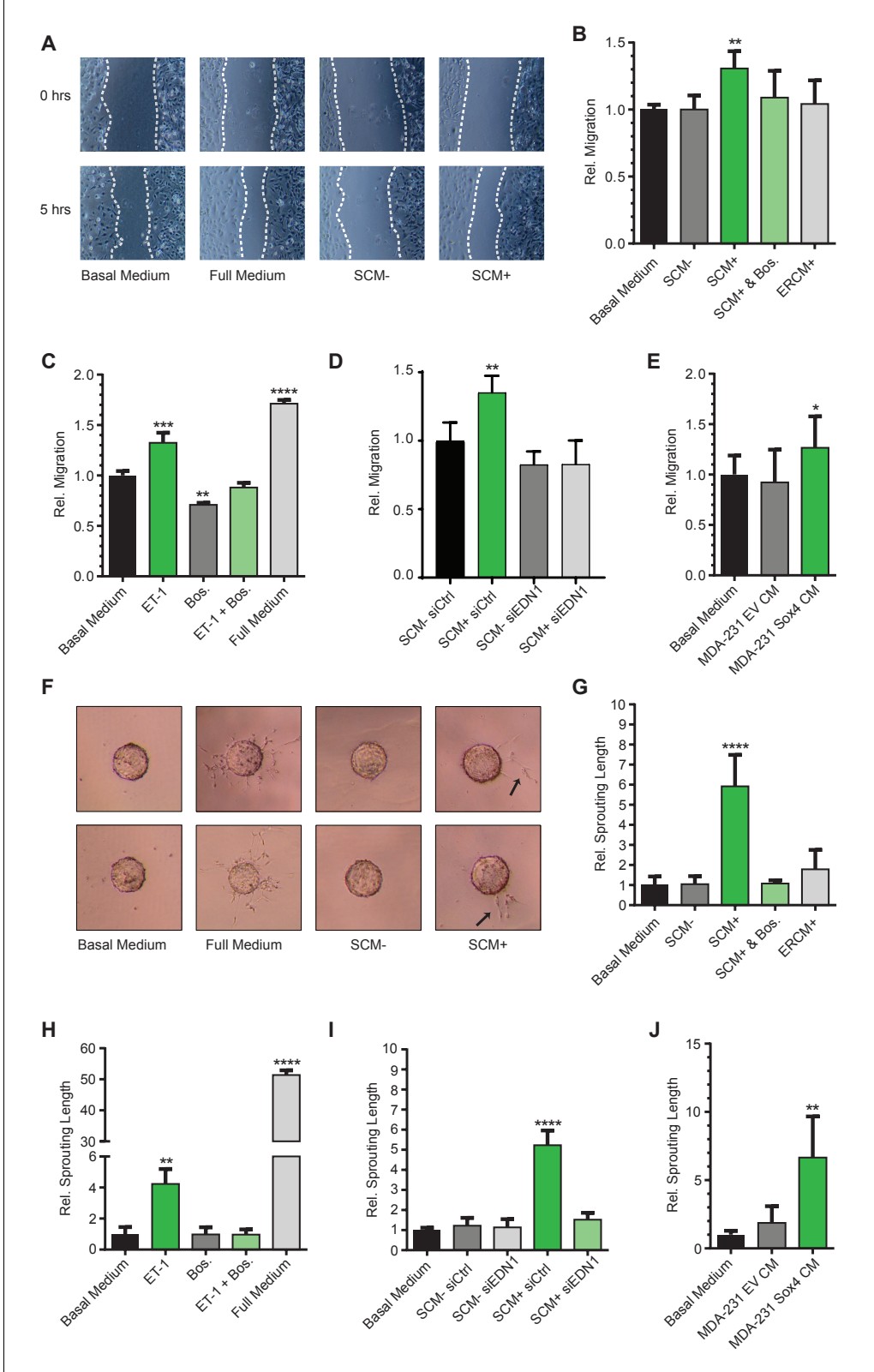

**Figure 4.** SOX4 mediated induction of ET-1 expression promotes tumor-induced angiogenesis in vitro. (**A**) Representative images of wound-healing assays performed on HMEC-1 cells, treated for 5 hr with basal media, full growth media or conditioned media from HMLE-S4 cells with or without overnight treatment with 4-OHT (SCM + and SCM-, respectively). Data from three independent biological replicates (**B**) Quantification of wound closure relative to basal media for HMEC-1 cells treated with conditioned media; SCM-, SCM+, SCM+ with Bosentan, and conditioned media from ER-control

*Figure 4 continued on next page*

*Figure 4 continued*

HMLE cells treated with 4-OHT (ERCM+). Data represented as mean ± SD of three independent biological replicates. (C) Quantification of wound closure in HMEC-1 cells treated with a synthetic ET-1 peptide, the ET-1 receptor antagonist Bosentan (Bos), ET-1 + Bos and full growth media represented as relative migration compared to basal media. Data represented as mean ± SD of four independent biological replicates. (D) Quantification of the wound healing assay in presence of SCM-/+generated from si*EDN1* or control siRNA treated HMLE cells. Results obtained from three independent biological replicates. (E) Quantification of the wound-healing assay in the presence of conditioned media from MDA-MB-231 cells stably expressing either an empty vector control (EV) or SOX4 construct. Results represented are relative to basal media. Data represented as mean ± SD of three independent biological replicates. (F) Representative images of sprouting assays performed with HMEC-1 cells in the presence of indicated media conditions. (G–J) Quantification of sprouting length in the indicated conditions relative to basal media. Data represented as mean ± SD of four independent biological replicates.

DOI: https://doi.org/10.7554/eLife.27706.011

The following figure supplement is available for figure 4:

**Figure supplement 1.** Conditioned media from conditionally activated ERSOX4 HMLE cells does not affect HMEC-1 cell proliferation.

DOI: https://doi.org/10.7554/eLife.27706.012

be quantified by fluorescence microscopy taking advantage of the presence of GFP-positive blood vessels characteristic of the Tg(fli:GFP) embryos used in these experiments.

ER-control and ERSOX4 HMLE cells were either pre-treated in vitro with 4-OHT for 24 hr or left untreated before injection into the yolk sac of zebrafish embryos (*Figure 5B*). Neovascularization was quantified by counting the number of blood vessels protruding from the SIV. A significant increase in sprouting from the SIV was observed in 4-OHT treated ERSOX4 HMLE cells compared to untreated ERSOX4 HMLE cells as well as both treated and untreated ER-control HMLE cells, indicating that conditional activation of SOX4 in HMLE cells can induce angiogenesis in vivo (*Figure 5B–D*). No significant changes were observed upon injection of wild-type HMLE cells (*Figure 5—figure supplement 1A*).

We next examined whether depletion of SOX4 could inhibit ectopic sprouting induced by pro-angiogenic MDA-MB-231 tumor cells. To this end, MDA-MB-231 cells stably expressing a shRNA targeting SOX4 (SOX4 KD) or a non-targeting shRNA were injected into zebrafish embryos. Quantification of neovascularization demonstrated that SOX4-depleted MDA-MB-231 cells (*Figure 5—figure supplement 1B*) were significantly impaired in their ability to induce ectopic sprouting from the SIV, as illustrated by a decrease in the number of sprouting vessels and an overall reduction in the percentage of fish with sprouting blood vessels (*Figure 5E–G*). In order to confirm that SOX4 pro-angiogenic effects *in vivo* are, in part, dependent of ET-1 expression, siRNA-mediated knockdown of *EDN1* was performed in HMLE cells 24 hr prior to SOX4 conditional activation, after which cells were injected into zebrafish embryos. Similar to our previous observations *in vitro*, depletion of ET-1 in SOX4-activated cells results in the reduction of angiogenic potential, shown by the decreased capacity to induce ectopic sprouting from the SIV (*Figure 5H–J*). Taken together, these results indicate that, similar to our in vitro observations, SOX4 controls tumor-induced angiogenesis *in vivo*, in an ET-1 dependent manner.

## SOX4 is required for ET-1 expression, tumor-vascularization and metastasis in a xenograft mouse model of breast cancer

To further study the role of SOX4 in tumor-induced angiogenesis in vivo, we made use of a xenograft mouse model of breast cancer, in which MDA-MB-231 cells are transplanted into the mammary fat-pad of immune-deficient mice. Luciferase expressing MDA-MB-231 cells were generated and lentivirally transduced with a SOX4 KD shRNA or a control-scrambled shRNA construct. Prior to injection, we validated the efficiency of *SOX4* knockdown by qRT-PCR. This revealed a significant reduction in *SOX4* expression in SOX4 KD MDA-MB-231 cells compared to scrambled control cells, which was accompanied by a significant decrease in *EDN1* expression (*Figure 6A–B*). Following validation, cells were transplanted into the mammary fat-pad and we subsequently monitored tumor growth up to six weeks post-implantation. Primary tumor growth was unaffected by SOX4 depletion and both scrambled control and SOX4 KD mice were sacrificed in the same week (*Figure 6C*). Analysis of primary tumors by immunohistochemistry (IHC) for SOX4, ET-1 and CD31 (PECAM-1) expression showed that SOX4 KD knockdown was maintained in vivo (*Figure 6D*). Moreover, quantitative analysis of ET-1 expression revealed a significant reduction in ET-1 IHC staining in

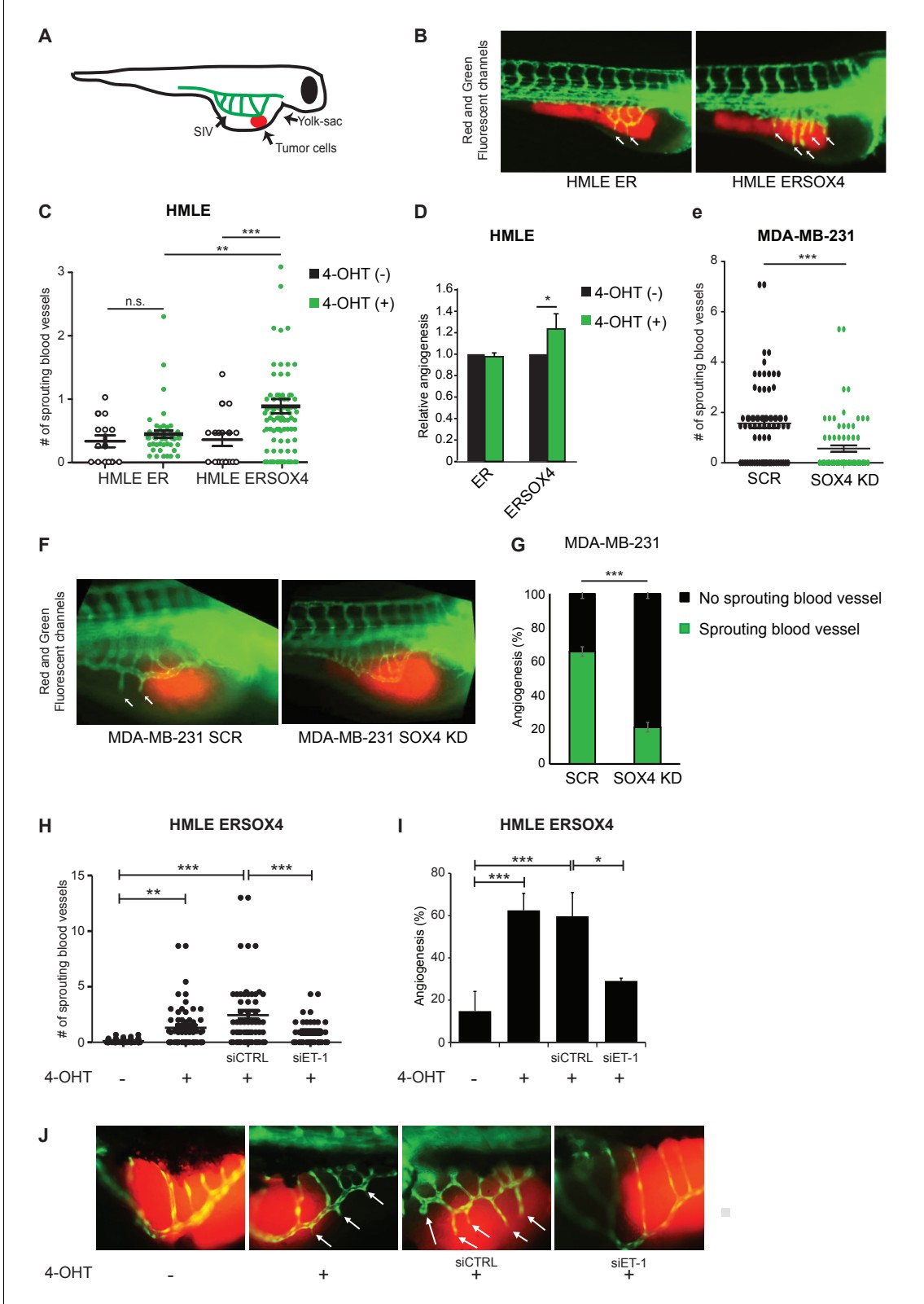

**Figure 5.** SOX4 controls tumor-induced angiogenesis in vivo in a zebrafish tumor-xenograft model. (**A**) Schematic representation of the zebrafish tumor-xenograft model, with the subintestinal vessels indicated in green (SIV) and tumor cells in red. (**B**) Representative images of zebrafish injected with ER-control and ERSOX4 HMLE cells treated with 4-OHT. Arrows indicate newly formed blood vessels. (**C**) Quantification of the number of ectopic sprouts observed per fish injected with ER-control and ERSOX4 HMLE cells. Individual data points and data represented as mean ± SD of three

*Figure 5 continued on next page*

*Figure 5 continued*

independent biological replicates (\*\*p-value<0.01, \*\*\*p-value<0.001; ANOVA). (D) Quantification of the relative increase in ectopic sprouting in fish injected with 4-OHT treated relative to untreated ER-control and ERSOX4 cells. Data represented as mean ± SD of three independent biological replicates (\* p-value<0.05, Student's t-test). (E) Quantification of the number of ectopic sprouts in fish injected with SCR control and SOX4 KD MDA-MB-231 cells. Results obtained from three independent biological replicates (\*\*\*p-value<0.001, Student's t-test). (F) Representative images of zebrafish injected with SCR control or SOX4 KD MDA-MB-231 cells. Arrows indicate newly formed blood vessels (G) Quantification of the number of fish injected with MDA-MB-231 cells with or without ectopic sprouting. Results obtained from three independent biological replicates (\*\*\*p-value<0.001, Student's t-test). (H) Quantification of the number of ectopic sprouts observed per fish injected with 4-OHT-treated ERSOX4 HMLE cells exposed with siRNA control or siRNA targeting ET-1. Individual data points and data represented as mean ± SD (\*p-value<0.05, ANOVA). (I) Quantification of the relative increase in ectopic sprouting in fish injected with 4-OHT treated ERSOX4 HMLE cells exposed with siRNA control or siRNA targeting ET-1. Data represented as mean ± SD of three independent biological replicates (\*p-value<0.05, \*\*\*p-value<0.001; ANOVA). (J) Representative images of zebrafish injected with 4-OHT-treated ERSOX4 HMLE cells exposed with siRNA control or siRNA targeting ET-1. Arrows indicate newly formed blood vessels.

DOI: https://doi.org/10.7554/eLife.27706.013

The following figure supplement is available for figure 5:

**Figure supplement 1.** Analysis of angiogenic potential of HMLE WT cells.
DOI: https://doi.org/10.7554/eLife.27706.014

SOX4 KD compared to scrambled control tumors (*Figure 6D–E*). We next assessed tumor vascularization by quantifying expression of the endothelial cell surface marker CD31. SOX4 KD tumors had significantly lower vascularization as illustrated by the reduction in total blood vessel area, blood vessel count and blood vessel size (*Figure 6F*). These findings demonstrate that SOX4 expression is required for tumor-associated ET-1 expression in vivo, and SOX4-depletion correlates with reduced tumor blood vessel density. Importantly, SOX4 depletion in this orthotopic xenograft model also significantly reduced the development of lung metastases (*Figure 6G*). To exclude the possibility that these results were simply caused by off-target effects of the shRNA, we performed an independent experiment using a second shRNA targeting SOX4 (*Figure 6—figure supplement 1*). This experiment confirmed our prior experiment, as mice transplanted with SOX4 shRNA2 MDA-MB-231 cells presented with a strong decrease in both metastatic outgrowth and vascularization. Here, there was also a modest effect on primary tumor growth, which may reflect better SOX4 knockdown in this experiment. Together, these findings demonstrate that SOX4 can control multiple aspects of breast tumor biology, and show for the first time that SOX4 may regulate tumor angiogenesis in vivo.

## SOX4 expression correlates with decreased survival, increased metastasis and elevated blood vessel density in patients with breast cancer

To evaluate whether SOX4 expression correlates with clinical parameters in breast cancer, we performed IHC analysis of nuclear SOX4 expression levels in a tissue microarray (TMA) representing a cohort of 452 breast cancer patient tumor samples from both invasive ductal (IDC) and lobular carcinomas (ILC) (*Figure 7A–B*). Nuclear SOX4 expression was found to significantly correlate with tumor size, histological grade and mitotic index, all parameters of poor-prognosis (*Figure 7—source data 1*, *Figure 7—source data 2*, *Figure 7—source data 3*, *Figure 7—source data 4*, *Figure 7—source data 5*). Accordingly, high levels of SOX4 expression significantly correlated with decreased overall survival in breast cancer patients, and this was also observed when patients were subdivided into the major subtypes ILC and IDC (*Figure 7C–E*). Stratification of patients with breast cancer according to the ER-status revealed that patients with high levels of SOX4 have a significantly worse prognosis in the ER-positive tumor group, and a trend towards poor-prognosis in the ER-negative tumor group (*Figure 7—figure supplement 1A*).

We next evaluated the correlation between nuclear SOX4 intensity and the metastatic status for those cases in the TMA cohort where this information was available. This revealed that the percentage of tumors with high SOX4 expression was observed to be significantly higher in those patients who develop metastasis as compared to patients who do not present with metastatic disease (p=0.016; *Figure 7—source data 5*). Moreover, analysis of SOX4 expression in 23 matched primary-tumor metastasis patient samples revealed that SOX4 expression in the primary tumor reflects the expression level observed in the metastasis (p=0.039; *Figure 7—source data 5*). These observations indicate that SOX4 expression is a clear marker of poor-prognosis in breast cancer and underscore

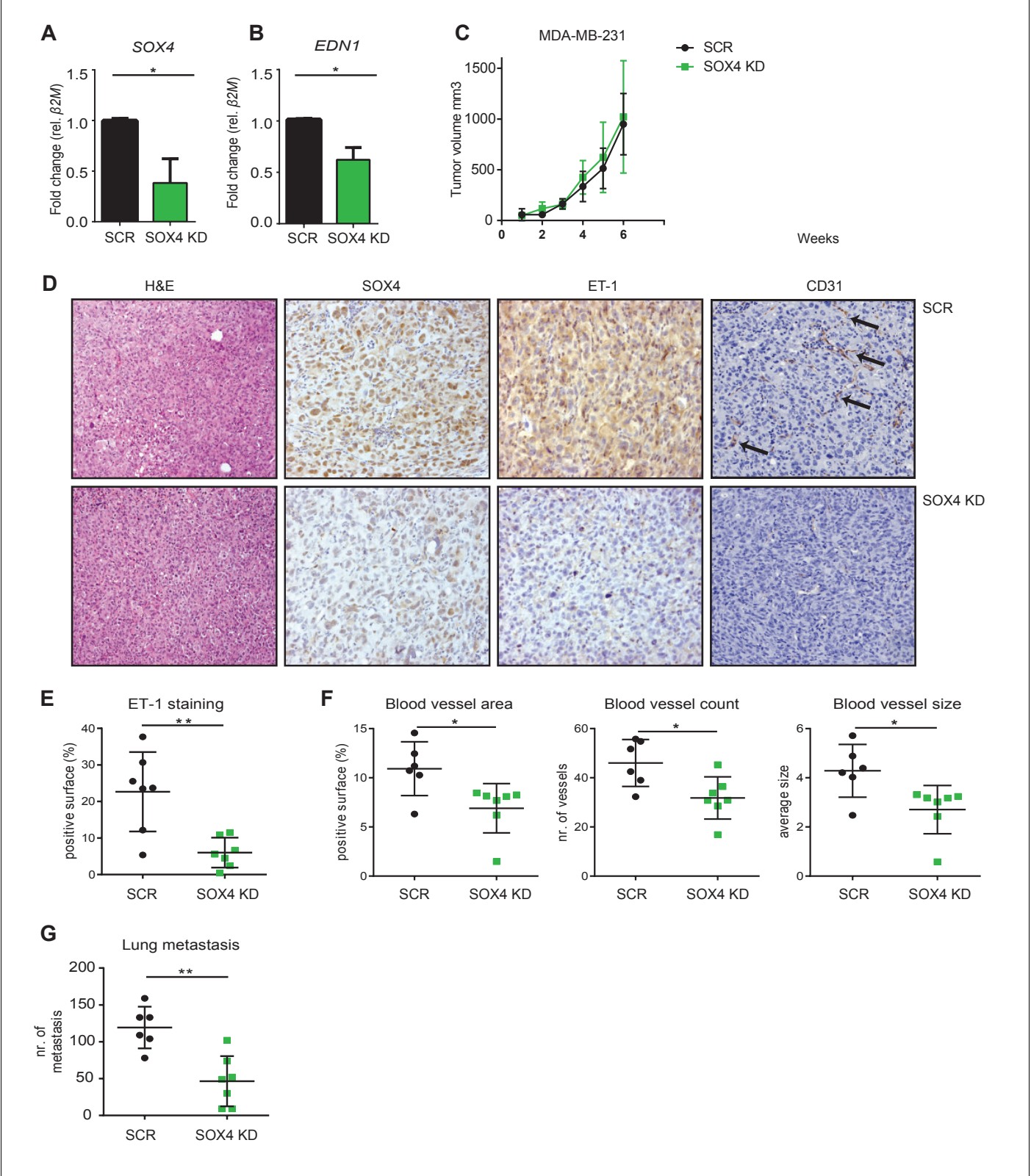

**Figure 6.** Depletion of SOX4 in MDA-MB-231 cells reduces ET-1 expression and decreases tumor vascularization and metastasis in a xenograft mouse model of breast cancer. (A–B). qRT-PCR quantification of *SOX4* and *EDN1* expression in luciferase-marked SCR control and SOX4 KD MDA-MB-231 cells. Data represented as mean ± SD of three independent biological replicates (*p-value<0.05, Student's t-test). (C) Growth curve of SOX4 KD and SCR control MDA-MB-231 tumors. Data represented as mean ± SD of three independent biological replicates (D) IHC analysis of scrambled control and

*Figure 6 continued on next page*

*Figure 6 continued*

SOX4 KD MDA-MB-231 primary tumors. Representative Hematoxylin and eosin stainings (H and E) and IHC stainings for SOX4, ET-1 and CD31 (PECAM-1) are shown. Arrows indicate blood vessels. (E) Quantification of ET-1 staining. Indicated is the total surface passing the filter threshold. Data represented as mean ± SD (**p-value<0.01, student t-test). (F) Quantitative analysis of the endothelial cell marker CD31 in SOX4 KD and control tumors. Images were analyzed by ImageJ software and total vessel area, blood vessel count and blood vessel size are indicated (*p-value<0.05, t-test). (G) Quantitative analysis of the number of visible lung metastases in SOX4 KD and SCR control MDA-MB-231 tumors. Data represented as mean ± SD (**p-value<0.01, t-test).

DOI: https://doi.org/10.7554/eLife.27706.015

The following figure supplement is available for figure 6:

**Figure supplement 1.** Depletion of SOX4 using a second ShRNA in MDA-MB-231 cells reduces Endothelin-1 expression and decreases tumor vascularization and metastasis in a xenograft mouse model of breast cancer.

DOI: https://doi.org/10.7554/eLife.27706.016

the important role of SOX4 in tumor-progression and metastasis observed in breast cancer model systems. Interestingly, no significant benefit to patient survival was observed for adjuvant radio- or chemotherapy-treated SOX4-high tumors as compared to untreated tumors, whereas a significant treatment response was observed for SOX4-low tumors (*Figure 7—figure supplement 1C*). These findings suggest that SOX4 may also contribute to therapy resistance.

To investigate whether SOX4 expression is also associated with increased angiogenesis in patients with human breast cancer, we examined CD34 expression (a marker for small and newly formed blood vessels) in 17 SOX4-low (SOX4$^{LO}$) and 17 SOX4-high (SOX4$^{HI}$) ductal carcinomas from our cohort (*Figure 7F*). Quantitative analysis of CD34-staining revealed that tumors expressing high levels of SOX4 had a significantly increased total blood vessel area and size, with an apparent but non-significant increase in the total number of blood vessels, thus suggesting that SOX4 can influence tumor-induced angiogenesis in human breast cancer (*Figure 7G*).

Taken together our integrative genome-wide analysis of the SOX4 transcriptional network has uncovered a novel role for SOX4 in tumor-induced angiogenesis in vitro and in vivo as well as in breast tumors, revealing a novel mechanism by which SOX4 may contribute to breast cancer metastasis (*Figure 7H*).

## Discussion

Using a combination of *in vitro* experiments utilizing multiple epithelial-derived breast cell lines, *in vivo* animal models, and analysis of patient tumor samples, we show that SOX4-mediated transcriptional regulation can affect breast cancer progression by promoting tumor-induced angiogenesis. SOX4 activation promotes ET-1 expression and controls the angiogenic behavior of endothelial cells both in vitro and in vivo, thus affecting tumor-progression in a non-cell autonomous manner. Finally, we demonstrate that elevated SOX4 expression correlates with increased blood vessel density in patient tumors and predicts poor-prognosis.

Despite observations of increased *SOX4* expression in a wide variety of tumors, effects can differ greatly between cancer-types resulting in cellular transformation, increased proliferation, EMT and metastasis, or even tumor-suppression (*Rhodes et al., 2004*; *Vervoort et al., 2013a*). The cell-type and context dependent effects underlying this differential SOX4 response are poorly understood. Our findings indicate that SOX4 acts as an opportunistic transcription factor directed by a pre-existing chromatin landscape, which upon recruitment to open chromatin further enhances the active chromatin-state. In addition to the epigenome, co-factors may determine SOX4 target gene selection as indicated by the co-occurrence of consensus-binding sites for AP-1, ETS-1 and SMAD3 in SOX4-bound sites. Cooperative binding appears to be a common theme in the SOX-family, as highlighted by SOX2 and OCT4 in embryonic stem cells, which results in the specific regulation of pluripotency genes including NANOG (*Kamachi et al., 2000*; *Sarkar and Hochedlinger, 2013*). These observations indicate that both the specific cell-type epigenome and the expression of co-factors are important in shaping the SOX4 transcriptional network, providing an explanation for the relatively minor degree of overlap in SOX4 target genes between cancer-types (*Vervoort et al., 2013a*). The context dependent nature of SOX4 transcriptional networks is of particular importance in heterogeneous diseases such as breast cancer. An important finding is that the SOX4 core-

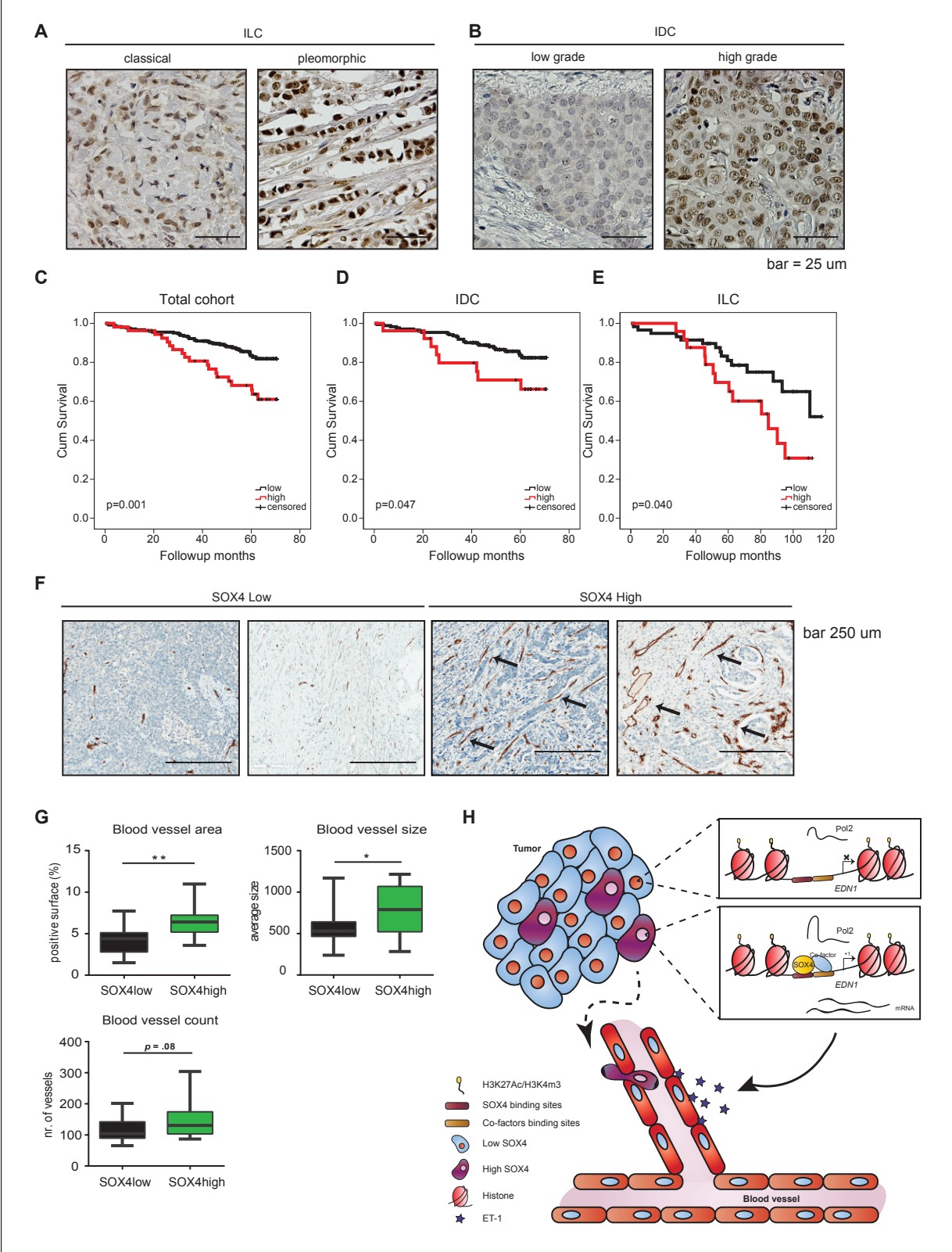

**Figure 7.** Immunohistochemical analysis of SOX4 expression in patients with human breast cancer reveals a positive correlation with decreased survival, increased metastasis and elevated blood vessel density. (A–B) Representative images of SOX4 expression in invasive lobular carcinoma (ILC, classical and pleomorphic) and invasive ductal carcinoma (IDC, low-grade tumor and high-grade tumor) cases in the tissue microarray (TMA) cohort. (C–E) Kaplan-Meier curves for cumulative (cum) survival for patients with high or low nuclear SOX4 expression for the total cohort, ILC and IDC breast

*Figure 7 continued on next page*

*Figure 7 continued*

cancers. Indicated p-values were calculated using the Log-rank test. (**F**) Representative images of CD34 staining in SOX4[HI] and SOX4[LO] tumors as identified previously in TMA analysis. Arrows indicate blood vessels. (**G**) Quantitative analysis of CD34 staining in 17 SOX4[HI] and 17 SOX4[LO] ductal carcinomas. (Student t-test, *p-value<0.05). (**H**) Schematic model for the pro-angiogenic function of SOX4 during breast cancer progression. The SOX4 transcription factor binds to regions of active/open chromatin and subsequently acts as positive regulator of many genes involved in tumorigenesis, such as EDN1. Upregulation of ET-1 contributes to the increase of neovascularization, which ultimately may facilitate tumor cell intravasation and growth factor accessibility.

DOI: https://doi.org/10.7554/eLife.27706.017

The following source data and figure supplements are available for figure 7:

**Source data 1.** Correlation of nuclear SOX4 expression with clinicopathological and molecular features of invasive breast cancer.
DOI: https://doi.org/10.7554/eLife.27706.020
**Source data 2.** Correlation of nuclear SOX4 expression with clinicopathological features of IDC and ILC.
DOI: https://doi.org/10.7554/eLife.27706.021
**Source data 3.** Clinicopathological characteristics of 452 breast cancer patients studied for the expression of SOX4.
DOI: https://doi.org/10.7554/eLife.27706.022
**Source data 4.** Clinicopathological characteristics of 27 patients with metastatic breast cancer studied for the expression of SOX4.
DOI: https://doi.org/10.7554/eLife.27706.023
**Source data 5.** Correlation of nuclear SOX4 expression with metastasis formation.
DOI: https://doi.org/10.7554/eLife.27706.024
**Figure supplement 1.** *SOX4* mRNA expression correlates with decreased patient survival in both ER positive and negative breast cancers.
DOI: https://doi.org/10.7554/eLife.27706.018
**Figure supplement 2.** Gene set enrichment analysis on RNA-seq data derived from conditional activation of SOX4 in HMLE cells.
DOI: https://doi.org/10.7554/eLife.27706.019

signature is strongly predictive for survival specifically in the Claudin[low] breast cancer subtype. Previous studies have identified distinct clusters of genes that distinguish Claudin[low] breast cancers from other subtypes. The nature of these gene clusters suggest that Claudin[low] breast tumors are characterized by cells that have undergone EMT, have extensive immune cell infiltration and also exhibit activation of the vasculature (*Prat and Perou, 2011*; *Taube et al., 2010*; *Harrell et al., 2014*; *Hennessy et al., 2009*). These are all processes that we observe can be regulated by SOX4 and is reminiscent of other EMT-transcription factors such as TWIST1 and SNAIL1 (*Prat et al., 2013*) that can induce an acquisition of Claudin[low] traits in HMLE cells. MDA-MB-231 is a Claudin[low] cell line (*Prat et al., 2013*), and thus our findings in the xenograft model may mimic the effects SOX4 has in Claudin[low] tumors.

SOX4 has been demonstrated to contribute to the TGF-β-induced EMT response (*Lourenço and Coffer, 2017*). Accordingly, we find that EMT and TGF-β associated genes are significantly enriched in our SOX4 target gene dataset (*Figure 7—figure supplement 2A–B*), although major EMT associated transcription factors *SNAI1*, *SNAI2* or *TWIST1* were not observed to be regulated (data not shown) (*Vervoort et al., 2013a*; *Tiwari et al., 2013*; *Zhang et al., 2012*). Despite a previous report that SOX4-mediated induction of EMT occurs through upregulation of EZH2 levels, we did not detect significant regulation of *EZH2* upon conditional SOX4 activation in HMLE cells (data not shown). This suggests that the direct transcriptional network of SOX4 is not dependent on the induction of *EZH2* in these cells. Nonetheless, it is possible that SOX4-mediated regulation of *EZH2* occurs upon prolonged activation of SOX4, thus contributing to EMT. These findings indicate that SOX4 may contribute to EMT and the TGF-β response in breast cancer on multiple levels.

We provide the first evidence of a potential role for SOX4 in tumor-induced angiogenesis. Primary tumor-growth was not consistently affected in vivo by SOX4 depletion, as we observed little consistent difference in tumor growth after SOX4 knockdown (*Figure 6C* and *Figure 6—figure supplement 1C*). However, we did observe that SOX4 KD tumors consistently exhibit fewer blood vessels (*Figure 6F* and *Figure 6—figure supplement 1E*). It should be noted that these experiments utilize the 'aggressive' MDA-MB-231 cell line where loss of SOX4 may not be sufficient to reduce primary tumor growth. However, even in this context the effects on metastasis and vascularization are clear (*Figure 6*, *Figure 6—figure supplement 1*). SOX4-induced tumor angiogenesis may also contribute to metastatic dissemination. We have shown that depletion of SOX4 leads to a reduced number of spontaneous lung metastases in a xenograft model, which correlated with decreased ET-1

expression and blood vessel density. It has been previously reported that gene signatures associated with vascular activation in Claudin[low] tumors were predictive for metastatic dissemination (*Harrell et al., 2014*). Our SOX4-regulated RNAseq data in HMLEs showed enrichment for one of these vascular signatures (*Figure 7—figure supplement 2C*) suggesting that SOX4 may drive tumor angiogenesis in Claudin[low] tumors by regulating these genes in addition to *EDN1*. This may contribute to the poor survival of patients with the SOX4 signature in patients with Claudin[low] tumors.

Thus, our study may signify a potential connection between SOX4-mediated alterations in the tumor-microvasculature and the ability of cells to metastasize. Based on our current data however we cannot determine the degree to which tumor angiogenesis contributes to the pro-metastatic effects of SOX4, since it is not possible to exclude other processes such as EMT and invasiveness. While previous studies have reported that SOX4 can directly modulate tumor cell transcription, our study has demonstrated for the first time that SOX4 can impact tumorigenesis in a non-cell-autonomous fashion.

In agreement with a role in tumor-progression, ET-1 expression has been demonstrated to positively correlate with tumor-grade and the formation of distant metastasis in breast cancer (*Wülfing et al., 2003*). In addition to promoting angiogenesis, SOX4 induced ET-1 may also promote tumor-progression in an autocrine manner by inducing EMT or for example by affecting T-cell infiltration, both of which have demonstrated in ovarian cancer (*Buckanovich et al., 2008*; *Rosanò et al., 2005*; *Rosanò et al., 2013*). SOX4-ET-1 induced autocrine and paracrine signaling may thus affect multiple tumor-promoting processes. It would be interesting to evaluate whether restoration of ET-1 expression is sufficient to restore the metastatic capacity of SOX4-depleted MDA-MB-231 cells in vivo. Since SOX4 does affect multiple processes that affect tumor progression we envision that ET-1 depletion would strongly reduce metastasis formation, but would most likely not abolish metastatic outgrowth completely.

In the last decade a number of anti-angiogenic therapies, mostly targeting the VEGF-receptor pathway, have been tested in clinical trials with limited success due to unwanted side-effects (*Jain, 2014*). These include the inhibition of effective delivery of anti-cancer therapeutics, but also the induction of pro-metastatic processes such as EMT, activated by nutrient deprivation and hypoxia (*Jain, 2014*). As a driver of both EMT and tumor-induced angiogenesis, SOX4 is an interesting therapeutic target in breast cancer, allowing for the simultaneous inhibition of multiple biological processes required for tumor metastasis. Alternatively, ETAR and ETBR receptor inhibitors such as Bosentan are currently used to treat pulmonary vascular hypertension, and based on our results could potentially be repurposed to specifically treat aggressive SOX4[HI] breast tumors (*Rosanò et al., 2013*). Although success of ET-1 receptor antagonists in the treatment of advanced cancers, including glioma and melanoma, has been limited, promising results were obtained in a number of ovarian cancer patients when combined with standard chemotherapeutics (*Rosanò et al., 2013*). Based on our findings, patients with SOX4[HI] tumors are more likely to profit from treatment with ET-1 receptor antagonists, thus arguing for the specific monitoring of this subgroup in future clinical studies. A caveat here is that considering the context-dependent nature of SOX4 transcriptional activity it is possible that *EDN1* regulation is also dependent on the expression of co-factors and on the presence of open chromatin. Despite our finding that *EDN1* was one of the few genes commonly regulated by SOX4 in the different breast epithelial cell lines used in this study, future work will have to uncover whether this regulation is conserved in all breast cancer cells and subtypes.

Taken together, our global-transcriptional analysis provides mechanistic insight into the contextual nature of SOX4 responses and uncovers a novel role for SOX4 in tumor-induced angiogenesis by direct regulation of ET-1 expression. The identification of this novel SOX4 controlled pathway underscores the validity of our approach and highlights the value of our dataset as a resource to uncover further novel functions for SOX4 in cancer and potentially to understand its role in developmental biology.

## Materials and methods

### Cell culture

Human Microvascular Endothelial Cells (HMEC-1) were obtained from the Center for Disease Control and Prevention, Atlanta USA (*Ades et al., 1992*) and maintained in MCDB 131 medium containing 10% FCS, 10 mM L-Glutamine, 100 U/ml penicillin and 100 µg/ml streptomycin (All from Life Technologies, USA), 50 nM hydrocortisone (Sigma-Aldrich, USA), and 10 ng/ml rhEGF (Invitrogen, USA) up to passage 27 at 37°C, 5% $CO_2$ and ambient oxygen level (20%). Immortalized human mammary epithelial cells (HMLE) were kindly provided by dr. R.A. Weinberg and were cultured in human mammary epithelial growth media (cc-3150, Lonza) as described previously (*Vervoort et al., 2013a*). MDA-MB-231 cells and HCC1954 cells were purchased from ATCC and cultured as described previously (*Bruna et al., 2012*). All cells were regularly checked for mycoplasma during the course of this study and only mycoplasma-negative cells have been used for the experiments described in this study.

### Lentiviral transduction

Lentiviral transduction of MDA-MB-231 and HMLE cells was performed as described previously (*Vervoort et al., 2013a*). Lentiviral constructs used were Control vector expressing shRNA control [(SHC002); Sigma-Aldrich, Missouri, USA] and shRNA targeting *SOX4* (TRCN0000018214, Sigma) for SOX4 KD1 used in *Figure 6*, and TRCN0000274152 for SOX4 KD2 used in *Figure 6—figure supplement 1*.

Knockdown of *EDN1* in HMLE cells was performed using 25 nM human *EDN1* siRNA (Thermo Scientific, ON-TARGET plus SMARTpool, L-016692-00-0010, Waltham, MA, USA) with Lipofectamine RNAiMAX reagents (Invitrogen), 24 hr before 4-OHT treatment.

### Statistics

Statistical analysis was performed using IBM SPSS Statistics version 21 (SPSS Inc., Chicago, IL, USA) for all patient data. Associations between categorical variables were examined using the Pearson's Chi-square test or Fisher's exact test when required. Survival analyses were performed using Kaplan-Meier and differences were analyzed using Log rank test. P-values<0.05 were considered to be statistically significant. Additional statistical analyses were performed using GraphPad software (v6), and statistical analyses used are indicated in the figure legends. Statistical analysis of in vitro data was performed using the analysis of variance (ANOVA) with a Post Hoc Dunnett's multiple comparisons test, unless indicated differently. Gene signature analysis was performed by calculating Spearman-rank correlations between centroids of the SOX4 signature and patient gene expression profiles. ChIP seq is represented as boxplots with 5–95% whiskers. Statistical comparison was performed with a two-tailed Mann-Whitney U test (****p<0.0001) for unpaired peak-sets and a two-tailed Wilcoxon-signed-rank test for paired peak-sets (****p<0.0001).

### Survival analysis

The Molecular Taxonomy of Breast Cancer International Consortium (METABRIC) cohort includes DNA copy number, whole gene expression and breast cancer-specific survival of 1980 patients (data is freely available and can be found on cBioPortal; accession number EGAS00000000083) (*Curtis et al., 2012*). Claudin^low subtyping was performed using transcriptomic data and the predictor developed by Prat et al. (*Prat et al., 2010*). Signature analysis was performed using the genes that were more than 1.5 log2 fold change differentially expressed between 4-OHT treated and untreated ERSOX4 HMLE in the transcriptomic characterization and bound in SOX4 ChIP-seq analysis of DOX treated and untreated HMLE-S4. Samples were assigned a Spearman correlation-based score calculated between the gene expression data of each tumour and each signature. Survival was analysed using Kaplan-Meier curves and significance assessed by log-rank test. Data were analysed using R 3.2.2 and a two-sided p<0.05 was considered significant.

### Patients

The study population was derived from the archives of the Departments of Pathology of the University Medical Center Utrecht, Utrecht, and the Radboud University Nijmegen Medical Center,

Nijmegen, The Netherlands. These comprised 452 cases of invasive breast cancer. Histological grade was assessed according to the modified Bloom and Richardson score (*Elston and Ellis, 1991*), and the mitotic activity index (MAI) was assessed as before (*van der Groep et al., 2006*). Other clinico-pathological characteristics are shown in Table S3-7. From representative donor paraffin blocks of the primary tumors, tissue microarrays were constructed as described by Vermeulen et al. (*Vermeulen et al., 2012b*). The use of anonymous or coded leftover material for scientific purposes is part of the standard treatment contract with patients in The Netherlands (*van Diest, 2002*). Ethical approval was therefore not required. Overall survival and treatment data were obtained from the Comprehensive Cancer Center of The Netherlands (Integraal Kankercentrum Nederland). Survival data were available of 295 out of 452 breast cancer cases, with a follow up of 72 months for the ductal and 120 months for the lobular breast cancer cases.

## Immunohistochemistry

Immunohistochemistry was carried out on 4 µm thick sections. Data on HER2, progesterone receptor (PR), estrogen receptor α (ERα), and E-cadherin were derived from Vermeulen et al. (*Vermeulen et al., 2012a*; *Vermeulen et al., 2012b*). After deparaffination and rehydration, endogenous peroxidase activity was blocked for 15 min in a buffer solution pH5.8 containing 0.3% hydrogen peroxide. After antigen retrieval, that is boiling for 20 min in 10 mM citrate pH6.0, a cooling period of 30 min preceded the primary antibody incubation. Primary antibodies against SOX4 (HPA029901, Sigma Aldrich) 1:50 were diluted in PBS containing 1% BSA and incubated overnight at 4°C. The signal was amplified using the Novolink kit (Leica, Rijswijk, The Netherlands) and developed with diaminobenzidine, followed by counterstaining with haematoxylin, dehydration in alcohol, and mounting. Appropriate negative and positive controls were used throughout.

All scoring was done blinded to patient characteristics and results of other stainings by three independent observers (SJV, JFV, PJvD). E-cadherin and HER2 stainings were scored using the DAKO/HER2 scoring system for membrane staining. Membranous scores 1+, 2+, and 3 + were considered positive. For HER2 only a score of 3 + was considered positive. ERα and PR were scored by estimating the percentage of positive tumor cells, considering cancers with more than 10% positive tumor nuclei as positive. Nuclear and cytoplasmic SOX4 staining intensity was scored as 0–3+, considering samples with 3 + staining intensity as positive. The Perou/Sorlie molecular classification was simulated by ER/PR/HER2 as before (*Kornegoor et al., 2012*).

The CD34 staining (Immunotech, QBEND10) was performed in a 1:800 dilution on a Ventana autostainer. For xenograft tumors in *Figure 6* immunohistochemistry was carried out on 4 µm thick sections and antigen retrieval was performed as described for patient material. Primary antibodies against SOX4 (HPA029901, Sigma Aldrich) CD31 (PECAM-1 Antibody (M-20), sc-1506) and ET-1 (N-8, SantaCruz, sc-21625) 1:50 were diluted in PBS containing 1% BSA and incubated overnight at 4°C. The signal was amplified using poly-HRP (Leica, Rijswijk, The Netherlands) and developed with diaminobenzidine, followed by counterstaining with haematoxylin, dehydration in alcohol, and mounting. For xenograft tumors in *Figure 6—figure supplement 1* immunohistochemistry was carried out on 4 µm thick sections and antigen retrieval was performed as described for patient material. Primary antibodies against SOX4 (HPA029901, Sigma Aldrich) Endomucin (ab106100 abcam) and ET-1 (N-8, SantaCruz, sc-21625) were diluted in PBS containing 1% BSA and incubated overnight at 4°C. Note that the Endomucin antibody was used to identify blood vessels as the PECAM-1 antibody was commercially not available. The signal was amplified using the Novolink kit (Leica, Rijswijk, The Netherlands) and developed with diaminobenzidine, followed by counterstaining with haematoxylin, dehydration in alcohol, and mounting.

To quantify blood vessels in tumors, ImageJ software was used. To quantify CD34 (patient tumor material) and CD31 (murine model) 3,3′-Diaminobenzidine (DAB) staining or Endomucin staining. Positive signal was isolated using the ImageJ colour deconvolution plug-in, followed by a fixed threshold background subtraction and automatic particle analysis, using a minimum particle size of 50 pixels.

## Migration assay

200,000 cells per well were plated 24 hr in advance in 24-well plate wells to form a confluent monolayer. Migration was assessed by making a single straight scratch in the HMEC-1 monolayer using a

20–200 µl pipette tip. Wells were washed with PBS to remove loose cells, and 0.5 ml test medium (basal-, full-, or conditioned medium) was added, followed by a 5 hr incubation at 37°C, 5% $CO_2$ and ambient oxygen level (20%). Images were recorded at t = 0 and t = 5 hr at similar locations, after which the scratch surface was determined using ImageJ software to determine scratch closure.

## Matrigel angiogenesis assay

The effect of conditioned medium on angiogenic activity of HMEC-1 cells was determined by seeding 2,000 cells (in 10 µl basal medium) on to 10 µl solidified Growth Factor Reduced Matrigel (Becton Dickinson, United Kingdom) in a µ-Slide Angiogenesis (Ibidi, Germany) in the presence of 50 µl test medium (basal-, full-, or conditioned medium). After an 18 hr incubation at 37°C, images were recorded using an Olympus BX53 microscope with a DP71 digital camera, and analyzed for total network length using the ImageJ Angiogenesis Analyzer plugin.

## Measurement of angiogenic sprouting

Angiogenic sprouting stimulation of conditioned medium was measured by seeding HMEC-1 cells on to Cytodex3 microcarrier beads (Sigma-Aldrich, USA) and embedding them in a 1:1:4 mixture of basal medium, test medium, and Growth Factor Reduced Matrigel (Becton Dickinson, USA). After solidification of the gels, 0.5 ml basal medium was added on top of the gels and incubated for 48 hr at 37°C, 5% $CO_2$ and ambient oxygen level (20%). Images were recorded using an Olympus CKX41 microscope with an SC30 digital camera. Sprout lengths were determined using ImageJ software.

## Measurement of proliferation

100,000 HMEC-1 cells were seeded in full HMEC-1 medium in 24-well plate wells 24 hr before the assay. 0.4% BSA was added to all test media except full HMEC-1 medium. Test media was added to the wells, and the cells were incubated for 48 hr at 37°C, 5% $CO_2$ and 20% $O_2$. Then, cells were washed with PBS and isolated using 0.5 ml 0.25% Trypsin-EDTA (Gibco, Invitrogen, USA) for 5 min at 37°C, flushed loose with 0.5 ml full HMEC-1 medium, and counted using a TC20 automated cell counter (Bio-Rad, USA).

## Chromatin Immunoprecipitation

Prior to chromatin immunoprecipitation (ChIP) and sequencing HMLE-S4 cells were grown in the absence or presence of 1 µg/ml of doxycycline for 16 hr, ChIP was performed as described previously (*van Boxtel et al., 2013*) using the following antibodies: 10 µg of SOX4 antibody (CS-129–100, Diagenode), 5 µg H3K4me3, H3K27ac, (ab8580, ab4729, Abcam), 5 µg anti-RBP1 (PB-7C2) antibody (Euromedex), and 5 µg H3K27me3 (39157, Active Motif). Truseq nano DNA sample preparation kit (Illumina) was used for End-repair, A-tailing, and ligation of sequence adaptors. Samples were subsequently PCR amplified after which the libraries were size selected to remove adapter dimers and select fragments in the 200–500 bp range. Barcoded libraries were sequenced on Illumina NextSeq500 sequencer (75 bp, single-end, Utrecht DNA sequencing facility).

## RNA sequencing

Total RNA was extracted from ER and ERSOX4 HMLE cells treated with 100 nM 4-OHT for 8 hr, using the RNAeasy kit (Qiagen, Germany). Sample preparation was performed using TruSeq stranded total RNA with ribo-zero globin sample preparation kit (Illumina, USA) and samples were sequenced 75 bp single-end on Illumina NextSeq500 (Utrecht DNA sequencing facility).

## NGS data analysis

BaseSpace software (Illumina) was used for sample demultiplexing, only reads with quality score of Q > 30 were subsequently used for further analysis. Differential gene expression analysis was performed using the Tophat2-cufflinks pipeline as described previously (*Trapnell et al., 2012*). Briefly, reads were mapped to the human reference genome (hg19) using TopHat v2.0.9 and using the reference gene annotation (hg19) as a guide transcripts were assembled using CuffLinks v2.2.1. Cuffdiff was used for differential gene analysis. Prior to differential gene analysis reads were quartile

normalized using the –library-norm quartile option and reads mapping to rRNA and tRNA were masked from the analysis (-M command). CummeRbund was used for quality assessment and figure generation. Cluster 3.0 and Java TreeView software version 1.1.6 were used for clustering and visualization of heatmaps. The DAVID gene ontology database (http://david.abcc.ncifcrf.gov/) and the ClueGO module in cytoscape were used for functional annotation of differentially expressed genes (*Huang et al., 2009*; *Bindea et al., 2009*).

ChIP-sequencing reads were mapped to the reference genome (hg19) with Bowtie 2.1.0 (*Langmead and Salzberg, 2012*) using default settings. Samtools version 0.1.19 was used for file conversions (SAM to BAM) and peaks were called using Cisgenome 2.0 using the input as a control (–e 150 -maxgap 200 –minlen 200) (*Jiang et al., 2010*). Mapped reads were extended according to the average fragment size and converted to TDF files with igvtools-2.3.36 and were visualized with IGV-2.3.34 (*Thorvaldsdóttir et al., 2013*). General manipulation of bed-files was performed using BEDtools v2.17.0 (*Quinlan and Hall, 2010*). Quantile normalization and quantification of ChIP-seq reads was performed as described previously (*Eijkelenboom et al., 2013*). HOMER software was used for motif discovery, peak annotation and the generation of histograms (*Heinz et al., 2010*).

## Analysis of gene expression signatures

The log fold change values for each of the different gene signatures were used to create a centroid which was then correlated to the z-score transformed centroid for the corresponding genes for each patient sample in METABRIC using the Spearman correlation as described in Bruna et al. (*Curtis et al., 2012*; *Bruna et al., 2012*). The 25th, median or 75th percentile or median Spearman correlation value was used as a cut-off for the Kaplan-Meier analysis of breast cancer-specific death and survival for subgroups defined by ER status.

## Zebrafish embryo maintenance, cell injection and imaging

Zebrafish maintenance and procedures were performed in the Faculty of Natural Sciences at Imperial College London (UK), in accordance to the UK Home office regulations (ASPA 1986). Zebrafish adult specimen were kept in a self-recirculating aquarium at an average temperature of 28°C with a 14 hr light 10 hr dark cycle. Adult specimens were fed twice a day on a diet of Hikari micropellets (Kyorin) and brine shrimp.

Zebrafish in vivo angiogenesis assays were performed on Tg(*fli*:GFP) embryos (*Lawson and Weinstein, 2002*). To facilitate visualization post-implantation, cell lines lacking intrinsic fluorescent signalling were pre-labeled with CM-DiI (Invitrogen, UK) according to the manufacturer's instructions. For injections, labeled cells were harvested to obtain a pellet without medium and then resuspended in 10 µl PBS. 1 day post-fertilization (1dpf) embryos were manually dechorionated and then anesthetized in a solution containing 0.003% tricane (Sigma, UK) for 10 min. Injections were performed using a Narishige microinjector with a 12 mm gage borosilicate pipette. During the procedure, embryos were kept on an injection mould composed of 3% agar in a solution of pre-warmed PBS (+Mg + Ca) with the addition of 0.003% tricane (Sigma, UK). Approximately, 150–200 cells were injected in the yolk sac of each embryo, in proximity of the sub-intestinal vessel complex. For individual experiments, approximately 25 embryos were injected per cell line. Injections were completed within 1 hr, following which the embryos were maintained in a solution of system water (chlorine deprived tap and distilled water mixture) with the addition of 0.0003% (v/v) methylene blue (antifungal) and 30 µg/ml N-phenylthiourea (Sigma) (PTU, averts melanin formation), and kept at a constant temperature of 28°C. Live imaging was performed on individual embryos 2 days post-injection (dpi) under a widefield fluorescent microscope (Olympus CKX41). Fish were anesthetized with the addition of 0.05% tricaine to their water, to prevent movement during the live imaging practice. All pictures were taken with Q Capture-Pro (QImaging) apparatus, with any alteration to the original picture performed via the use of Image J.

## Orthotopic tumor xenograft model

Mammary fat-pad transplantations in female RAG2$^{-/-}$;IL-2Rγc$^{-/-}$ immunodeficient mice using approximately $1 \times 10^6$ luciferase-expressing MDA-MB-231 cells and bioluminescence imaging were performed as described previously (*Ivanova et al., 2013*). Tumour growth was measured weekly using

a digital caliper (VWR, Radnor, PA, USA) and mice were sacrificed when tumor volume exceeded 1000 mm$^2$ or visible metastases were detected by BLI. All animal experiments were approved by the Utrecht University Animal Experimental Committee (DEC-Utrecht no. 2013.III.02.020). Immediately after resection, the tumors were fixed in neutral buffered formalin and paraffin embedded. Immunohistochemistry was performed on 4 μm thick sequential sections, as described above. Macroscopic lung metastases were counted, ET-1 and CD31 staining was quantified using ImageJ as described above.

## qPCR

For qPCR analysis, mRNA was isolated using the RNAeasy kit (Qiagen, Germany) according to the manufactures' protocol after which cDNA was generated using the iScript cDNA synthesis kit (Bio-Rad). Subsequently, real-time quantification was performed on a MyiQ Single-Color Real-Time PCR Detection System (Bio-Rad, USA) using SYBR Green Supermix (Bio-Rad, USA) for cDNA amplification according to the manufacturer's protocol.

| | | | |
|---|---|---|---|
| FW-SOX4 | ACCGCACGCCAAGCTCATCC | FW-LMO4 | AAAAGCAGACCATGGTGAATCC |
| RV-SOX4 | GTCCGCGCCTTGTACAGCGA | RV-LMO4 | GCTGTGCCAATAGCTGTCCA |
| FW-B2M | CCAGCAGAGAATGGAAAGTC | FW-SMARCC1 | CTGTTGCAGCCAACATCCAC |
| RV-B2M | CCAGCAGAGAATGGAAAGTC | RV-SMARCC1 | GGGGGATACATGCCGTTAGG |
| FW-*EDN1* | TTGAGATCTGAGGAACCCGC | FW-ECE1 | TGGCACCCACAACTGCAAAT |
| RV-*EDN1* | GCTCAGCGCCTAAGACTGTT | RV-ECE1 | GTACGTCGACATCTGGGTCC |
| FW-MARCKSL1 | GGCTACAGAGCCATCCACTC | FW-MMP10 | GACACAGTTTGGCTCATGCCTACCC |
| RV-MARCKSL1 | TGACCTCACAAGGACAGCAC | RV-MMP10 | TTGGTGCCTGATGCATCTTCTGTCC |
| FW-DBN1 | GGAGTTCTTCCAGGGTGTCG | FW-TEAD2 | CCGCTCGAGAGTGTGGACGT |
| RV-DBN1 | GTCTTCTGGTAGGTGGTGCC | RV-TEAD2 | AGGTCCGCCCAGAACTTGACCAG |
| FW-MARCKS | TGCCCAGTTCTCCAAGACCGC | FW-FZD7 | ACAGAGGCCCAGGGACGAAAGC |
| RV-MARCKS | GCCATTCTCCTGTCCGTTCGCT | RV-FZD7 | CTCTCCCAACCGCCTCGTCGCA |

## Luciferase assays

For the luciferase assay, HEK293T cells at 50% confluency were transfected in 24-well plates with 0.1 μg EDN-1 promoter luciferase reporter or mutated versions thereof, with 0.1 μg of pcDNA3 empty vector, pcDNA3- flag-SOX4, pcDNA3 flag-SOX4 DN (aminoacids 1–135), and 0.02 μg pRL-TK Renilla (Promega) as a transfection control. The cells were lysed in 50 μl passive lysis buffer 3 days post-transfection, the soluble fraction was subsequently assayed for luciferase activity with a Dual-Luciferase Reporter Assay System (Promega, USA).

## Oligonucleotide Pull-Down Assay

Oligonucleotide pull-down assays were performed as described previously (*Vervoort et al., 2013a*). Briefly, cell extracts of HEK239T cells transfected with either an empty vector control pcDNA or flag-SOX4 pcDNA were incubated with biotinylated double-stranded oligonucleotide probes, which were precoupled to streptavidin beads. Binding sites matching the consensus SOX4 binding sequence (*Vervoort et al., 2013a*) or the identified ET-1 promoter regions were used. After incubation, with the biotinylated probes the samples were washed in low-salt buffer (150 mM NaCl) and subsequently analyzed by western blot using an anti-flag antibody.

| | |
|---|---|
| ET-1 −70BPfw | GGTGACTAATAACACAATAACATTGTCTGGGGCTGGAATAAA |
| ET-1 −70BPrv | TTTATTCCAGCCCCAGACAATGTTATTGTGTTATTAGTCACC |
| ET-1 −350BPfw | ATTCCCCGCACACAACAATACAATCTATTTAAACTGTGGCTCA |
| ET-1 −350BPrv | TGAGCCACAGTTTAAATAGATTGTATTGTTGTGTGCGGGGAAT |
| ET-1 −70BPmutfw | GGTGACTAATAACACAATACTCCCACCTGGGGCTGGAATAAA |

*Continued on next page*

| | |
|---|---|
| ET-1 −70BPmutrv | TTTATTCCAGCCCCAGGTGGGAGTATTGTGTTATTAGTCACC |
| ET-1 −350BPmutfw | ATTCCCCGCACACCCTGGCACAATCTATTTAAACTGTGGCTCA |
| ET-1 −350BPmutrv | TGAGCCACAGTTTAAATAGATTGTGCCAGGGTGTGCGGGGAAT |

## ELISA

HMLE-ERSOX4 cells and HMLE-ER cells were seeded at 50.000 cell/well in a 6-well plates and were subsequently allowed to adhere overnight. The cells were subsequently treated or left untreated for 8 hr with 100 nM 4-hydroxytamoxifen (4-OHT)after which the media was replaced with HMEC-1 bare media for overnight conditioning in the presence or absence of 4-OHT. The next day the media was collected and cell-debris was removed by using a 20 µM filter. Samples were subsequently diluted 10x and analyzed for ET-1 expression using the Endothelin-1 Quantikine ELISA Kit (DET100, R and D systems) according to the manufacturer's protocol.

## Acknowledgements

The authors would like to thank Dr. RA Weinberg for the kind gift of HMLE cells, Hisham Mohammed and Jason Carroll for assistance with RIME assays. We would also like to thank J. van Loosdregt, R van Boxtel and J Beekman for helpful discussions. OGdJ is supported by The Netherlands Institute for Regenerative Medicine (NIRM, grant No. FES0908), MCV is supported by The Netherlands Organization for Scientific Research (Vidi grant 016.096.359) SJV, MGR, CLF and ARL are funded by the Dutch Cancer Foundation (UU-2013–5801 and UU-2015–7838). EFWL and LB are supported by grants from Breast Cancer Campaign (2012NovemberPhD016) and Cancer Research UK (A12011). Eric W-F Lam's work is supported by MRC (MR/N012097/1), CRUK (C37/A12011;C37/A18784), Breast Cancer Now (2012MayPR070; 2012NovPhD016), the Cancer Research UK Imperial Centre, Imperial ECMC and NIHR Imperial BRC. Laura Bella was supported by Breast Cancer Now (2012NovPhD016).

## Additional information

### Funding

| Funder | Grant reference number | Author |
|---|---|---|
| KWF Kankerbestrijding | KWF UU 2013-5801 | Stephin J. Vervoort<br>Ana Rita Lourenço |
| Dutch Cancer Foundation | UU-2013–5801 | Stephin J Vervoort<br>M Guy Roukens<br>Cynthia L Frederiks<br>Ana Rita Lourenço |
| The Netherlands Institute for Regenerative Medicine | FES0908 | Olivier G de Jong |
| KWF Kankerbestrijding | KWF UU 2015-7838 | Guy Roukens<br>Cynthia L Frederiks |
| Fundação para a Ciência e a Tecnologia | | Ana Rita Lourenço<br>José L Sandoval |
| Breast Cancer Campaign | 2012NovemberPhD016 | Eric W-F Lam<br>Laura Bella |
| Cancer Research UK | A12011 | Eric W-F Lam<br>Laura Bella |
| Breast Cancer Now | 2012NovPhD016 | Laura Bella |
| Cancer Research UK | | Alejandra Bruna<br>Carlos Caldas |
| Breast Cancer Campaign | | Alejandra Bruna<br>Carlos Caldas |

| The Netherlands Organization for Scientific Research | 016.096.359 | Marianne C Verhaar |
|---|---|---|
| MRC | MR/N012097/1 | Eric W-F Lam |
| Cancer Research UK | C37/A12011 | Eric W-F Lam |
| Breast Cancer Now | 2012MayPR070 | Eric W-F Lam |
| Cancer Research UK Imperial Centre | | Eric W-F Lam |
| Imperial ECMC and NIHR Imperial BRC | | Eric W-F Lam |
| Cancer Research UK | C37/A18784 | Eric W-F Lam |
| Breast Cancer Now | 2012NovPhD016 | Eric W-F Lam |
| Dutch Cancer Foundation | UU-2015–7838 | Stephin J Vervoort<br>M Guy Roukens<br>Cynthia L Frederiks<br>Ana Rita Lourenço |

The funders had no role in study design, data collection and interpretation, or the decision to submit the work for publication.

## Author contributions

Stephin J Vervoort, Cynthia L Frederiks, Conceptualization, Data curation, Formal analysis, Validation, Methodology, Writing—original draft; Olivier G de Jong, Jeroen F Vermeulen, Ana Rita Lourenço, Laura Bella, Ana Tufegdzic Vidakovic, José L Sandoval, Cathy Moelans, Miranda van Amersfoort, Margaret J Dallman, Edward Nieuwenhuis, Elsken van der Wall, Patrick Derksen, Paul van Diest, Michal Mokry, Conceptualization, Data curation, Formal analysis, Validation, Methodology; M Guy Roukens, Conceptualization, Data curation, Supervision, Funding acquisition, Methodology, Writing—review and editing; Alejandra Bruna, Carlos Caldas, Marianne C Verhaar, Eric W-F Lam, Conceptualization, Resources, Supervision, Investigation, Methodology, Writing—review and editing; Paul J Coffer, Conceptualization, Resources, Supervision, Funding acquisition, Investigation, Methodology, Writing—original draft, Project administration, Writing—review and editing

## Author ORCIDs

Paul J Coffer http://orcid.org/0000-0001-8775-1939

## Ethics

Human subjects: The use of anonymous or coded leftover material for scientific purposes is part of the standard treatment contract with patients in The Netherlands. Ethical approval was therefore not required.
Animal experimentation: All experiments were performed in accordance to international guidelines and approved by Experimental Animal Committee Utrecht (DEC-Utrecht, University Utrecht, Utrecht, The Netherlands).

## Decision letter and Author response

Decision letter https://doi.org/10.7554/eLife.27706.035
Author response https://doi.org/10.7554/eLife.27706.036

# Additional files

## Supplementary files

• Transparent reporting form
DOI: https://doi.org/10.7554/eLife.27706.027

## Data availability

All ChIPseq data and RNAseq data has been deposited to GEO (GSE104761).

The following datasets were generated:

| Author(s) | Year | Dataset title | Dataset URL | Database and Identifier |
|---|---|---|---|---|
| SJ Vervoort, MG Roukens, PJ Coffer | 2018 | ChIP-seq HMLE vs HMLES4 | https://www.ncbi.nlm.nih.gov/geo/query/acc.cgi?acc=GSE104761 | Gene Expression Omnibus, GSE104761 |
| SJ Vervoort, MG Roukens, PJ Coffer | 2018 | ChIP-seq MDA-MB-231 | https://www.ncbi.nlm.nih.gov/geo/query/acc.cgi?acc=GSE104761 | Gene Expression Omnibus, GSE104761 |
| SJ Vervoort, MG Roukens, PJ Coffer | 2018 | ChIP-seq HC1954 | https://www.ncbi.nlm.nih.gov/geo/query/acc.cgi?acc=GSE104761 | Gene Expression Omnibus, GSE104761 |
| SJ Vervoort, MG Roukens, PJ Coffer | 2018 | RNA-seq HMLE vs HMLES4 | https://www.ncbi.nlm.nih.gov/geo/query/acc.cgi?acc=GSE104761 | Gene Expression Omnibus, GSE104761 |

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
