## [Decision Letter]

Thank you for submitting your article "Global transcriptional analysis identifies a novel role for SOX4 in tumor-induced angiogenesis" for consideration by *eLife*. Your article has been reviewed by three peer reviewers, and the evaluation has been overseen by Michael Green as the Reviewing Editor and Kevin Struhl as the Senior Editor. The reviewers have opted to remain anonymous.

The reviewers have discussed the reviews with one another and the Reviewing Editor has drafted this decision to help you prepare a revised submission.

Summary:

Vervoort et al. report on the identification of a SOX4 breast cancer transcriptional gene set associated with angiogenesis. Using a conditional system where SOX4 is induced and immunoprecipitated, the authors identify potential transcriptional targets of SOX4 identified in a human mammary epithelial line and two breast cancer lines. Gene Set Enrichment Analysis reveals Gene Ontology categories such as migration, extracellular matrix, and angiogenesis. The authors focus on one target gene – endothelin-1 – and provide data consistent with direct transcriptional targeting by SOX4. Consistent with an angiogenic role for SOX4, conditioned media from SOX4-over-expressing mammary epithelial cells exhibit modestly enhanced angiogenic activity, which is partially dependent on endothelial 1. Implantation of the SOX4-over-expressing mammary epithelial line and SOX4-depleted MDA-MB-231 breast cancer line in zebrafish reveal a pro-angiogenic role for SOX4 in this system. SOX4 depletion is also found to reduce tumor angiogenesis and orthotopic metastasis in an MDA-MB-231 xenograft model. Finally, human tumor tissue-microarray analyses reveal a positive association between SOX4 protein levels and metastatic relapse and tumor blood vessel density. These observations are consistent with angiogenesis promotion constituting one phenotype downstream of SOX4 and reveal endothelin-1 to represent a potential direct target and downstream mediator of SOX4's role in angiogenesis and metastasis.

Essential revisions:

1) In multiple experiments, the authors employ a single RNAi or shRNA to draw conclusions. One cannot exclude such observations representing off-target effects. Ideally, all experiments should be done with an independent RNAi. However, I'm willing to accept the use of a second independent RNAi in a single study that I deem most germane to the conclusion of the paper: Figure 6. The authors should generate a second SOX4 hairpin expressing breast cancer line and conduct a primary tumor growth and metastasis assay. In these same cells, when the authors validate the knockdown, they should validate reduction in expression of endothelin-1. In Figure 6A-B, the impact of SOX4 depletion on endothelin-1 is not that robust. A second hairpin with potentially better knockdown may reveal a greater effect. Moreover, Figure 6A-B appears to represent technical replicates. Biological replicates are essential. Moreover, the authors should generate pre-mRNA primers to ensure that the effects of SOX4 depletion also impact pre-mRNA transcript levels rather than solely SOX4 mRNA levels.

2) The endothelin-1 mutant reporter and its wildtype variant should be transfected into a breast cancer line (such as the cells generated above in (1)-along with its 2 SOX4 depleted lines) to demonstrate that endogenous SOX4 depletion leads to reduced endothelin-1 transcriptional reporter expression. This is essential, since the over-expression studies are non-physiological.

3) Citing of literature on prior papers on SOX4 and transcriptional targets regulating invasion.

4) When SOX4 was originally implicated in loss-of-function studies as a promoter of breast cancer metastasis, its associated phenotype implicated was invasiveness. Many subsequent papers have shown SOX4 to promote invasion in a number of systems. The authors should be commended for identifying an additional phenotype downstream of SOX4, but the Introduction does not discuss invasion as a key aspect of SOX4's biology in cancer. They mention EMT being associated, but in past work, enhanced invasiveness rather than EMT has been repeatedly implicated. This past work is consistent with the authors' identification of migration, ECM, and lack of observations of genes such as *SNAI1, SNAI2*, or *TWIST1* being modulated. The authors should cite the many prior papers by multiple groups on SOX4 describing its roles in invasiveness, including studies that identify transcriptional targets of SOX4 as pro-invasive genes and promoters of metastasis, including studies in breast cancer.

5) One major issue is whether the pro-angiogenic function of SOX4 is a conserved function of SOX4 in biology. Figure 1 shows that SOX4 mainly binds to regions of open chromatin, and then Figure 1—figure supplement 1H-I show that there is very little overlap between ChIP-seq data from HMLE, MB-231 and HCC1954. Therefore, the authors concluded that SOX4-target genes are very context dependent. Most of data on the connection of SOX4 with *EDN1* and angiogenesis is based on overexpressing SOX4. Only endogenous SOX4 knockdown was performed in MB-231 cells with a single shRNA and showed very limited data (no SOX4 or *EDN1* protein expression by western). A large panel of cancer cell lines should be examined to determine whether there is a correlation between SOX4 and *EDN1*. Figure 3— figure supplement 2C attempts to show that there is a correlation between SOX4 and EDN-1 in TCGA data. However, R=0.139 suggests that 80% samples show no correlation, further weakening the argument that SOX4 and EDN-1 is correlated at the RNA level. Therefore, data presented in the paper are not sufficient to conclude whether the link between SOX4 and *EDN1* pays a major role in angiogenesis.

6) Another major issue is whether the observed phenotypes of SOX4 in cancer is due to defects in angiogenesis. Figure 6 is the only data to look at the effects of knocking down endogenous SOX4 on tumor formation and metastasis. Given the major role of angiogenesis in promoting primary tumor formation, it is surprising that SOX4 knockdown has no effect on primary tumor growth despite increases in vessel area and vessel count. As the authors discussed in the Introduction, previous studies show that SOX4 induces EMT in various tumor cells. Therefore, the observed decrease in lung metastasis could be due to an EMT defect. Similarly, the correlation between SOX4 upregulation and poor prognosis in human breast cancer in Figure 7 could also be due to an EMT defect, instead of angiogenesis.

7) Figure 4A-4E show very mild effects of SCM from SOX4-overexpressing cells. In Figure 4D, since siEDN alone without SCM already affected cell migration, it is unclear whether siEDN with SCM indeed regulates the effect of SOX4 or independent of SOX4.

8) The authors convincingly show that Sox4 induces an angiogenic phenotype on endothelial cells, and that it is primarily mediated by endothelin-1. However, in the in vivo studies, the link between this angiogenic phenotype and metastatic dissemination is merely correlative. Sox4 knock down in MDA-MB-231 cells has been already shown to reduce metastatic dissemination through a cell migration and invasion defect. Is restoration of ET-1 in the context of Sox-4 knockdown sufficient to restore metastatic ability, as the authors suggest?

---

## [Author Response]

Essential revisions:1) In multiple experiments, the authors employ a single RNAi or shRNA to draw conclusions. One cannot exclude such observations representing off-target effects. Ideally, all experiments should be done with an independent RNAi. However, I'm willing to accept the use of a second independent RNAi in a single study that I deem most germane to the conclusion of the paper: Figure 6. The authors should generate a second SOX4 hairpin expressing breast cancer line and conduct a primary tumor growth and metastasis assay. In these same cells, when the authors validate the knockdown, they should validate reduction in expression of endothelin-1.

We have performed the experiment suggested using a second short hairpin RNA against SOX4 in MDA-MB-231 cells. As shown in Figure 6—figure supplement 1A-B the new cell line that has been generated to express a second shRNA against SOX4 also shows a strong reduction in expression endothelin-1. These cells were subsequently transplanted into NSG-mice and primary tumor growth and metastasis formation was assessed. In this experiment SOX4 reduction led to a modest decrease in primary tumor growth (Figure 6—figure supplement 1C). Importantly, SOX4 reduction induced a robust and significant decrease in the number of metastases in the lungs (Figure 6—figure supplement 1F). Supporting our previous data, the SOX4 depletion also led to a decrease in vascularization (Figure 6—figure supplement 1E). These important data confirm our earlier findings in Figure 6 and support a role for SOX4 in the regulation of endothelin-1, tumor-induced angiogenesis and metastasis.

In Figure 6A-B, the impact of SOX4 depletion on endothelin-1 is not that robust. A second hairpin with potentially better knockdown may reveal a greater effect. Moreover, Figure 6A-B appears to represent technical replicates. Biological replicates are essential.

The experiments shown in Figure 6A-B are in fact biological replicates. We apologize for the misunderstanding which may have stemmed from the fact that there are no error bars for the scrambled control cell line. These graphs represent relative expression for *SOX4* and *EDN1* similar to the other qRT-PCR data in the paper. We have set the control cell line (scrambled shRNA) to 1.0 in three independent experiments, and have compared the relative expression of the two genes in the knockout line.

Additionally, the same cell lines have been analyzed as technical triplicates in each experiment. We have now also shown in the Figure 6—figure supplement 1A-Bthat a second shRNA also induces a reduction of SOX4 and *EDN1* in an entirely independent cell line.

Moreover, the authors should generate pre-mRNA primers to ensure that the effects of SOX4 depletion also impact pre-mRNA transcript levels rather than solely SOX4 mRNA levels.

SOX4 is a single exon gene and there is no pre-mRNA that differs from a mature spliced mRNA.

2) The endothelin-1 mutant reporter and its wildtype variant should be transfected into a breast cancer line (such as the cells generated above in (1)-along with its 2 SOX4 depleted lines) to demonstrate that endogenous SOX4 depletion leads to reduced endothelin-1 transcriptional reporter expression. This is essential, since the over-expression studies are non-physiological.

We have now performed this experiment multiple times under a variety of conditions but we do not see a consistent difference between the activation of the endothelin-1 reporter in the MDA-MB-231 scrambled control cell line and the two SOX4 depleted cell lines. There could be a variety of explanations for this. By using an artificial transcriptional system such as a luciferase reporter construct, the reporter may not be appropriately regulated in MDA-231 cells. It is possible that low levels of SOX4 remain despite the knockdown and these are sufficient to activate the reporter under these artificial conditions. We would argue that we have shown that endogenous SOX4 binds the endothelin-1 promoter in MDA-MB-231 cells (Figure 3B) as the ChIP for MDA-MB-231 cells has been performed for endogenous SOX4. Secondly, we have now also shown with two different shRNAs that *EDN1* expression is also strongly reduced in the SOX4 shRNA lines (Figure 6A-B and Figure 6—figure supplement 1A-B). These data strongly support our hypothesis that endogenous SOX4 regulates *EDN1* expression. While we appreciate the reviewer’s suggestion, we don’t think that transfection of the reporter construct is a more physiological approach.

3) When SOX4 was originally implicated in loss-of-function studies as a promoter of breast cancer metastasis, its associated phenotype implicated was invasiveness. Many subsequent papers have shown Sox4 to promote invasion in a number of systems. The authors should be commended for identifying an additional phenotype downstream of SOX4, but the Introduction does not discuss invasion as a key aspect of SOX4's biology in cancer. They mention EMT being associated, but in past work, enhanced invasiveness rather than EMT has been repeatedly implicated. This past work is consistent with the authors' identification of migration, ECM, and lack of observations of genes such as SNAI1, SNAI2, or TWIST1 being modulated. The authors should cite the many prior papers by multiple groups on SOX4 describing its roles in invasiveness, including studies that identify transcriptional targets of SOX4 as pro-invasive genes and promoters of metastasis, including studies in breast cancer.

We appreciate the reviewer’s suggestion and have now added this discussion to the manuscript Introduction (fourth paragraph).

4) One major issue is whether the pro-angiogenic function of SOX4 is a conserved function of SOX4 in biology. Figure 1 shows that SOX4 mainly binds to regions of open chromatin, and then Figure 1—figure supplement 1H-I show that there is very little overlap between ChIP-seq data from HMLE, MB-231 and HCC1954. Therefore, the authors concluded that SOX4-target genes are very context dependent. Most of data on the connection of SOX4 with EDN1 and angiogenesis is based on overexpressing SOX4. Only endogenous SOX4 knockdown was performed in MB-231 cells with a single shRNA and showed very limited data (no SOX4 or EDN1 protein expression by western). A large panel of cancer cell lines should be examined to determine whether there is a correlation between SOX4 and EDN1. Figure 3—figure supplement 2C attempts to show that there is a correlation between SOX4 and EDN-1 in TCGA data. However, R=0.139 suggests that 80% samples show no correlation, further weakening the argument that SOX4 and EDN-1 is correlated at the RNA level. Therefore, data presented in the paper are not sufficient to conclude whether the link between SOX4 and EDN1 pays a major role in angiogenesis.

We have included multiple lines of evidence in diverse biological systems that suggest that the pro-angiogenic function of SOX4 is a conserved function in mammary tumor progression. In Figure 4 we have shown this effect of different mammary cell lines in in vitroassays. These same cell lines have also been used in zebrafish and in mice to show the angiogenic effects of SOX4 in vivo(Figure 5, Figure 6F, Figure 6—figure supplement 1E). In this revised version of the manuscript we show angiogenic effects of the overexpression system but also of the knockdown system with two independent shRNAs in separate experiments. Finally, in an entirely unrelated set of human mammary tumor samples we have shown that high SOX4 protein expression correlates with more vascularization (Figure 7G).

As to the concerns regarding the conserved nature of the regulation of *EDN1* by SOX4 we have also generated (additional) knockout cell lines for SOX4 in MDA-MB-231 cells (Figure 6—figure supplement 1A-B) and for HCC1954 (Figure 3—figure supplement 2C). In both these lines downregulation of endogenous SOX4 leads to a downregulation of *EDN1*. Since *EDN1* was one of a limited set of genes that is commonly bound by SOX4 in HMLE, MDA-MB-231 and HCC1954 cells, these data support the idea that EDN1 regulation by SOX4 is a conserved function.

We have also analyzed co-expression of SOX4 and *EDN1* in other databases. While there is a significant correlation (p-value) in both the TCGA and METABRIC database the correlation coefficient is not very high (TCGA R= 0.14; METABRIC R= 0.10). We have removed the previous Figure 3—figure supplement 2C from the revised manuscript. The absence of a strong correlation in these datasets could be due to a myriad of reasons. SOX4 gene expression does not always correlate with protein expression and we have previously demonstrated that the SOX4 protein can be stabilized by Syntenin (Beekman et al., 2012). Moreover, *EDN1* expression can be regulated by a variety of developmental and environmental cues. The *EDN1* promoter has been demonstrated to be responsive to TGFβ, Hypoxia and FOXO and NFKb mediated regulation (Stow et al., 2011). Importantly, these large scale RNA-seq efforts such as TCGA and METABRIC indiscriminately sample both tumour and stromal components including vasculature, which makes it difficult to interpret which cell type gave rise to the correlation.

Collectively our data shows that SOX4 can control *EDN1* expression in a variety of cellular systems and contexts. *EDN1* belongs to a subset of genes that was bound by SOX4 in three different mammary epithelial cell lines. Therefore it would appear likely that *EDN1* is one of the ‘core’ SOX4 target genes. However large-scale datasets do not provide clarity on the interrelated expression of these two genes. Therefore it is conceivable that regulation of *EDN1* by SOX4 is context dependent, similar to the majority of SOX4 target genes, which potentially rely on the expression of co-factors, distinct transcription factors, signaling input or a permissive chromatin state. We have included this notion in the Discussion (sixth paragraph). Importantly we have “softened” the conclusion that SOX4 is a major regulator of tumor-induced angiogenesis, but rather can be a regulator depending on context.

5) Another major issue is whether the observed phenotypes of SOX4 in cancer is due to defects in angiogenesis. Figure 6 is the only data to look at the effects of knocking down endogenous SOX4 on tumor formation and metastasis. Given the major role of angiogenesis in promoting primary tumor formation, it is surprising that SOX4 knockdown has no effect on primary tumor growth despite increases in vessel area and vessel count.

Indeed in the initial mouse experiment we were also surprised that the knockdown of SOX4 in MDA-MB-231 cells did not affect primary tumor growth despite its clear effects on vascularization. However, these cells are highly aggressive and perhaps are therefore better able to compensate for the loss of SOX4 in the primary tumor. In our second mouse experiment (Figure 6—figure supplement 1C) loss of SOX4 does lead to a modest reduction in primary tumor size, and this correlates with better SOX4 knockdown. Angiogenesis is a process that is executed when pro-angiogenic factors are able to overcome the effects of anti-angiogenic factors. These are in vivo experiments with tumors that exhibit natural variability in gene expression. Therefore it is conceivable that loss of SOX4-induced expression of pro-angiogenic factors is compensated by the expression of other pro-angiogenic factors in some of the experiments. We have now discussed this in the fourth paragraph of the Discussion section.

As the authors discussed in the Introduction, previous studies show that SOX4 induces EMT in various tumor cells. Therefore, the observed decrease in lung metastasis could be due to an EMT defect. Similarly, the correlation between SOX4 upregulation and poor prognosis in human breast cancer in Figure 7 could also be due to an EMT defect, instead of angiogenesis.

We appreciate the reviewer’s comment and clearly also show that SOX4 affects multiple processes, such as migration and cell cycle in our ChIP-seq data (Figure 3—figure supplement 1). The main message of our study is that SOX4 *can* regulate tumor angiogenesis. This is an important finding since angiogenesis impacts multiple stages of tumor development. In contrast to suggesting that SOX4 mediates metastasis solely by inducing tumor angiogenesis we would rather argue that SOX4 has such a major impact on (breast) cancer because it affects multiple pro-oncogenic processes. One of these is tumor angiogenesis, which we define here for the first time. This will be of major interest to the fields of breast cancer, EMT and SOX4.

It is extremely difficult, if not impossible, to dissect how much the angiogenic defect contributes to metastatic dissemination since EMT and angiogenesis are linked through a variety of mechanisms. For example, it has been demonstrated that EMT itself can induce the upregulation of VEGF and that this is an integral component of the EMT-mediated pro-tumorigenic effects (Fantozzi et al., 2014). Moreover, *EDN1* inhibition (either pharmacologically or using genetic approaches) would still be able to affect both processes as *EDN1* has been associated with EMT before (Rosanò et al., 2013). Alternatively, using general pharmacological inhibitors of angiogenesis (such as avastin) would not really add anything as it wouldn’t be clear whether any effect one sees has any relationship with SOX4.

We have now stressed more clearly in the fourth paragraph of the Discussion that we cannot distinguish how SOX4-induced tumor angiogenesis contributes to the pro-oncogenic effects of SOX4.

6) Figure 4A-4E show very mild effects of SCM from SOX4-overexpressing cells. In Figure 4D, since siEDN alone without SCM already affected cell migration, it is unclear whether siEDN with SCM indeed regulates the effect of SOX4 or independent of SOX4.

We have now repeated these experiments and confirmed that reduction of *EDN1* levels by siRNA in the HMLE ERSOX4 cell line inhibits migration of endothelial cells (Figure 4D revised version). We observe that there is also a reduction of cell migration also in the SCM- + si*EDN1* condition. It should be noted that we use conditioned medium from ERSOX4 HMLE cells without the addition of 4-OHT, but not “without SCM” as mentioned by the reviewers. This is of relevance since we find in our qRT-PCR analyses that there is already baseline *EDN1* expression in the HMLE ERSOX4 cells without 4-OHT (Figure 4—figure supplement 1B), which likely reflects low-level of leakiness of the inducible construct. Therefore, by depleting *EDN1* in this condition one would expect that this also leads to a reduction in endothelial cell migration.

Our new analyses further show that in the improved experimental setup loss of *EDN1* expression is comparable in HMLE ERSOX4 with and without 4-OHT (SCM- vs SCM+). This is also reflected in a similar degree by which the migration of endothelial cells is inhibited. Thus, collectively these data suggest that loss of *EDN1* can inhibit SOX4-dependent stimulation of endothelial migration.

*7) The authors convincingly show that Sox4 induces an angiogenic phenotype on endothelial cells, and that it is primarily mediated by endothelin-1. However, in the* in vivo *studies, the link between this angiogenic phenotype and metastatic dissemination is merely correlative. Sox4 knock down in MDA-MB-231 cells has been already shown to reduce metastatic dissemination through a cell migration and invasion defect. Is restoration of ET-1 in the context of Sox-4 knockdown sufficient to restore metastatic ability, as the authors suggest?*

As discussed with the editor, while an interesting experiment, this is beyond the scope of the current manuscript. Since we don’t want to suggest that *EDN1 alone* is responsible for metastasis we have now addressed this point in the Discussion (fourth paragraph).